# Uncoupled evolution of the Polycomb system and deep origin of non-canonical PRC1

Bastiaan de Potter [1,4], Maximilian W. D. Raas[1,2], Michael F. Seidl [1], C. Peter Verrijzer[3,5] & Berend Snel [1,5 ✉]

Polycomb group proteins, as part of the Polycomb repressive complexes, are essential in gene repression through chromatin compaction by canonical PRC1, mono-ubiquitylation of histone H2A by non-canonical PRC1 and tri-methylation of histone H3K27 by PRC2. Despite prevalent models emphasizing tight functional coupling between PRC1 and PRC2, it remains unclear whether this paradigm indeed reflects the evolution and functioning of these complexes. Here, we conduct a comprehensive analysis of the presence or absence of cPRC1, nPRC1 and PRC2 across the entire eukaryotic tree of life, and find that both complexes were present in the Last Eukaryotic Common Ancestor (LECA). Strikingly, ~42% of organisms contain only PRC1 or PRC2, showing that their evolution since LECA is largely uncoupled. The identification of ncPRC1-defining subunits in unicellular relatives of animals and fungi suggests ncPRC1 originated before cPRC1, and we propose a scenario for the evolution of cPRC1 from ncPRC1. Together, our results suggest that crosstalk between these complexes is a secondary development in evolution.

[1] Theoretical Biology and Bioinformatics, Department of Biology, Science Faculty, Utrecht University, Utrecht, Netherlands. [2] Hubrecht institute, Royal Netherlands Academy of Arts and Sciences, Utrecht, Netherlands. [3] Department of Biochemistry, Erasmus University Medical Center, Rotterdam, Netherlands. [4] Present address: Hubrecht institute, Royal Netherlands Academy of Arts and Sciences, Utrecht, Netherlands. [5] These authors contributed equally: C. Peter Verrijzer, Berend Snel. ✉email: b.snel@uu.nl

Packaging of the eukaryotic genome into chromatin is crucial for its maintenance, replication and expression. The nucleosome, comprising 147 bp of DNA wrapped in a left-handed superhelical turn around a histone octamer, is the fundamental repeating unit of chromatin. Histones and additional structural chromatin proteins impede access of DNA-binding proteins, such as transcription factors, thereby generating a fundamental level of gene regulation. Consequently, modulation of chromatin structure through post-translational histone modifications, nucleosome re-positioning and higher-order folding are central to eukaryotic gene regulation[1]. Polycomb group (PcG) proteins are conserved chromatin regulators that repress gene transcription to maintain cellular identity in plants and animals[2–4]. The *PcG* genes were first identified in *Drosophila* as repressors that are crucial to prevent mis-expression of homeotic (*Hox*) genes in inappropriate parts of the body[2–4]. This function is reflected by the name Polycomb, which refers to the extra sex combs appearing on the second and third pair of legs of male flies, due to de-repression of the *sex combs reduced* gene. Subsequent research showed that, in addition to silencing of *Hox* genes, PcG proteins are involved in the regulation of cell proliferation, stem cell pluripotency and oncogenesis[1,5–7].

PcG proteins function as part of two main classes of multi-protein complexes, named Polycomb Repressive Complex (PRC) 1 and 2, which change chromatin structure across multiple scales[2,4,6–9]. The PRC2 methyltransferases mediate mono-, di- and tri-methylation of H3K27 (H3K27me3), whereas the PRC1s mediate chromatin compaction and mono-ubiquitylation of H2A (H2Aub1) on Lys 118, 119 or 120 in *Drosophila*, mammals or *Arabidopsis*, respectively (Fig. 1A). The catalytic core of PRC2 comprises the methyltransferase EZH associated with EED, SUZ12 and RBBP (or their paralogs)[10,11]. EZH proteins contain a catalytic SET domain that is responsible for methylating H3K27, while EED is a WD40 repeat protein that allosterically activates EZH upon binding to H3K27me3, thus creating a positive feedback loop[10–12]. The C-terminal VEFS-box of SUZ12 further stabilizes the interaction between EED and EZH. The WD40 repeat protein RBBP, enhances chromatin association through direct binding to histones. Beyond its canonical core subunits, PRC2 engages in interactions with an array of accessory proteins. These accessory subunits, including but not limited to JARID2, AEBP2, and PCL, associate with the PRC2 core, leading to the formation of distinct subassemblies. These subassemblies perform specialized functions, allowing PRC2 to adapt to varying cellular contexts[4,10–18].

The PRC1 class can be subdivided in canonical (cPRC1) and non-canonical PRC1 (ncPRC1)[4,7–9,19–21]. Both are assembled around a RING-RING heterodimer, which in animals comprise a RING1 protein paired with a PcG Finger (PCGF) protein. Notably, plant RING1 and PCGF orthologs can form homo- as well as heterodimers[4]. RING1 and PCGF proteins share a similar structural organization characterized by an N-terminal zf-RING domain (usually known as RING finger domain) and a C-terminal RAWUL domain (Fig. 1B). cPRC1 is defined by the association of a Chromobox protein (CBX) and Polyhomeotic homolog (PHC) subunit, which are absent in ncPRC1[20]. Additionally, SCM is a PcG protein that is more loosely associated with cPRC1. The presence of a RYBP/YAF subunit defines the ncPRC1s (also known as variant PRC1 complexes), which through association with additional partner proteins form multiple variant complexes. Pertinently, H2Aub1 is deposited primarily by ncPRC1, rather than cPRC1[19,20,22–25].

The relative importance of the different PRC activities and their order of recruitment to chromatin remains controversial[1,6,7,9,24,26–35]. While there is compelling evidence for the essential function of H3K27me3 for Polycomb repression[25,36,37], the role of H2Aub1 is more ambiguous[22,24–35,38]. Studies in mouse embryonic stem cells have indicated that H2Aub1 is central to gene repression by the Polycomb system[24,27,30]. However, other studies in mouse embryonic stem cells revealed redundancy between H2Aub1 and H3K27me3[31,33,34]. Although H2Aub1 is essential for animal viability, mutational studies established that PRC1 rather than its enzymatic activity is essential for transcriptional repression of canonical Polycomb target genes[32,38–40]. Moreover, cPRC1 mediates long-range chromatin interactions via polymerization of the SAM domains within PH, which is independent of H2Aub1[41–43]. Thus, it appears that chromatin compaction is essential for Polycomb repression in animals, while H2Aub1 stimulates this process but is not absolutely essential. Intriguingly, while the core subunits RING1 and PCGF of PRC1 are well-conserved in plants, the nc- and cPRC1 accessory subunits appear to be absent. This implies an alternate scenario for the potential formation of higher-order Polycomb structures in

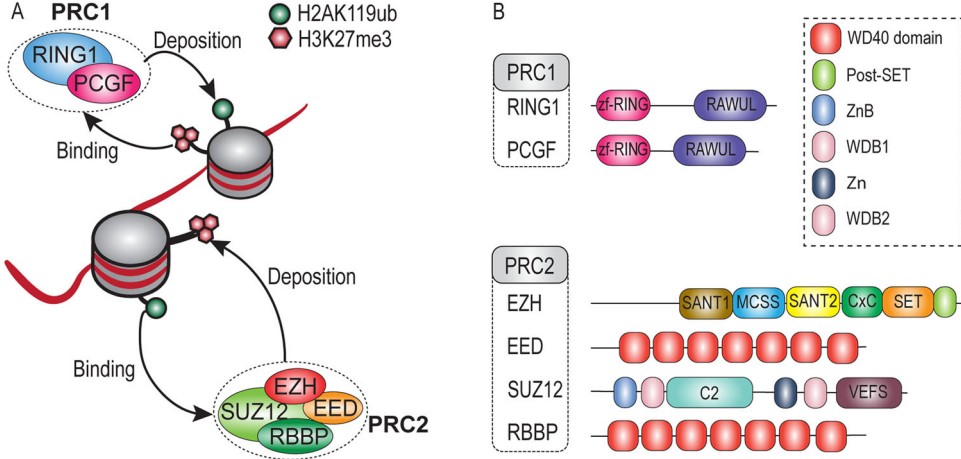

**Fig. 1 Schematic representation of the core subunits of PRC1 and PRC2 and their enzymatic activities. A** Overview of histone modifications by PRC1 and PRC2, and cross-talk via recognition of each other's histone marks **B** Schematic representation of the core subunits of PRC1 and PRC2. Shown are the major conserved domains for each protein based on structural information in humans. In EZH, we did not depict the additional SBD, EBD, BAM, SAL, and SRM domain. RAWUL RING finger- and WD40-associated ubiquitin-like, MCSS Motif connecting SANT1 and SANT2, SANT SWI3, ADA2, N-CoR and TFIIIB DNA-binding domain, CxC Cysteine-rich domain, SET Su(var) 3–9, enhancer of zeste, trithorax domain, WD WD-40 domain, WDB WD-40 binding domain, Zn Zn-finger region, VEFS VRN2-EMF2-FIS2-SU(Z)12.

plants[4]. Thus, while the molecular analysis of mechanisms of Polycomb repression have been based mainly on animal cells, it is important to realize that the Polycomb system might function differently in plants and non-metazoan organisms. The Polycomb system plays a crucial role in the developmental regulation of plants and in their response to environmental changes[44–46]. Moreover, there is accumulating evidence that the Polycomb system has diverse roles in genome regulation in fungi[47,48].

The binding of CBX proteins in cPRC1 to H3K27me3, and conversely, the binding of PRC2-accessory proteins to H2Aub1, provides cross-talk between PRC1 and PRC2[12,27,49]. These observations prompted the formulation of hierarchical recruitment models through "writing" and "reading" of histone modifications by PRC1 and PRC2 (Fig. 1A)[1,6,7,9,49]. At odds with such scenarios, there are accumulating examples of largely uncoupled gene repression by PRC1 and PRC2[31–35,38,50,51]. Moreover, it is becoming increasingly clear that in mammals, as previously established in *Drosophila*, DNA-binding plays a major role in the genomic targeting of PRCs[1,3,7,9,12]. In summary, PRC recruitment involves a combination of cooperative- and redundant mechanisms that may differ depending on the cellular- or genomic context. However, the functional- and evolutionary coupling between PRC1 and PRC2 remains an unresolved question.

Our knowledge of the function and evolution of the Polycomb system is primarily based on studies on animals and plants. This has limited our understanding of its role in developmental gene regulation in other eukaryotic lineages[52–55]. Recent research has started to expand the diversity of eukaryotes in which Polycomb repression is analyzed. For example, studies in ciliates have shown that PRC2-mediated trimethylation of histones is involved in the silencing of transposons[56–58]. In addition, a phylogenetic screening has shown that PRC2 is conserved across major eukaryotic groups and is likely present in the last eukaryotic common ancestor (LECA)[59]. However, PRC1 has not been studied as extensively in non-animal lineages, and therefore, whether PRC1 and PRC2 co-evolved remains unresolved.

In the present study we used highly sensitive and comprehensive phylogenetic profiling to analyze the evolutionary history of PRC1 and PRC2 across the available scope of genomes covering the eukaryotic tree of life. Our analysis suggests that both PRC1 and PRC2 were present in LECA. Importantly, we found that while their intra-complex subunits evolved cohesively, PRC1 and PRC2 evolve independently. Furthermore, we discovered previously unreported ncPRC1-defining subunits in the relatives of animals and fungi, suggesting that the differentiation of ncPRC1 occurred before cPRC1. These findings provide a foundation for future research on the biological functions of these proteins in a wider range of eukaryotes and in areas where PRC1 is understudied.

## Results
### Conservation of PRC1 subunits throughout eukaryotes traces its presence to LECA.
To study the evolution of PRC1 beyond animals and plants, we traced the presence of PRC1 core subunits, RING1 and PCGF, in a diverse collection of 178 eukaryotes that covers all known major eukaryotic groups (SI Data and Methods and Supplementary Data 1). Both RING1 and PCGF consist of an N-terminal zf-RING domain and a C-terminal RAWUL domain (Fig. 1B). These protein domains are the structural and functional units that are responsible for PRC1 functionality. As such, the conservation of specific protein domains across different species can be a strong indication that the protein is performing the same or similar functions in those species. To study the domain conservation of RING1 and PCGF proteins across our major eukaryotic groups, we utilized Pfam (v.35) domain profile

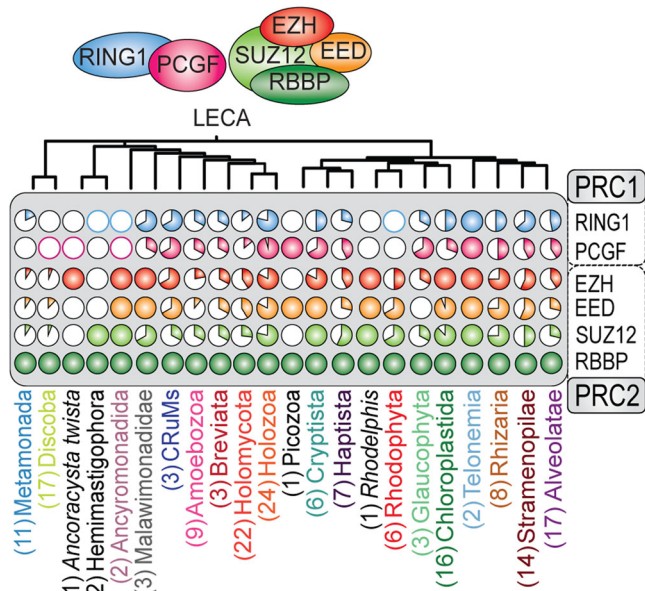

**Fig. 2 The evolutionary history of the core subunits of PRC1 and PRC2 can be traced back to LECA (Fig. S1A–E).** Shown is a summary of the phylogenetic profiles of PRC1 and PRC2 in all major eukaryotic groups included in our study (SI Fig. S2). Detected presences are presented as fractions of the species included in that group illustrated as a pie chart (Supplementary Data 9). For groups in which we only detected putative orthologs, only the outer line of the pie chart was colored.

hidden-Markov models (HMMs) to annotate the sequences included in the phylogenetic trees[60]. These profiles, however sometimes failed to detect zf-RING or RAWUL domains in protein sequences that clustered in potential orthologous groups, while their presence was evident after manual inspection of sequence alignments combined with AlphaFold2 structural predictions[61–63]. We therefore manually curated HMM profiles for the zf-RING, and RAWUL domain, based on our sequence alignments and structural predictions[61,63]. A detailed description of our approach and curated profiles are presented in SI Data and Methods and Supplementary Data 2.

This approach enabled us to identify 69 and 67 species with high-confidence orthologs for RING1 and PCGF, respectively. We considered an ortholog of high-confidence if it clustered monophyletically with the other sequences and contained both the zf-RING and RAWUL domain as predicted by our sequence models and AlphaFold2 (Fig. 2, SI Fig. S1A, B, Fig. S2). Additionally, we detected 15 species with a putative RING1- and ten species with putative PCGF orthologs, containing a zf-RING-, but lacking a recognizable RAWUL domain (SI Fig. S1A, B, Fig. S2). The phylogenetic trees of RING1 and PCGF contained multiple groups of sequences from species that are part of all major eukaryotic groups in which the deep duplication nodes displayed large species overlap, which is a defining feature of orthologous groups in a gene phylogeny. The subsequent speciation node can thus be identified with high reliability as a LECA presence (Figs. 2, 3A, S1A, B). The increased sensitivity of our approach is illustrated by the detection of a previously unknown RING1 ortholog in the diatom (Stramenopile) *Phaeodactylum tricornutum*. In addition, we were able to detect a PCGF ortholog in *Arabidopsis thaliana* that was missed in a recent study[64], most likely because the canonical Pfam model (v.35) is not sensitive enough to detect the RAWUL domain. Importantly, we identified RING1 and PCGF orthologs in all unicellular eukaryotes that are currently placed proximal to the root of the eukaryotic tree of life, such as Discoba,

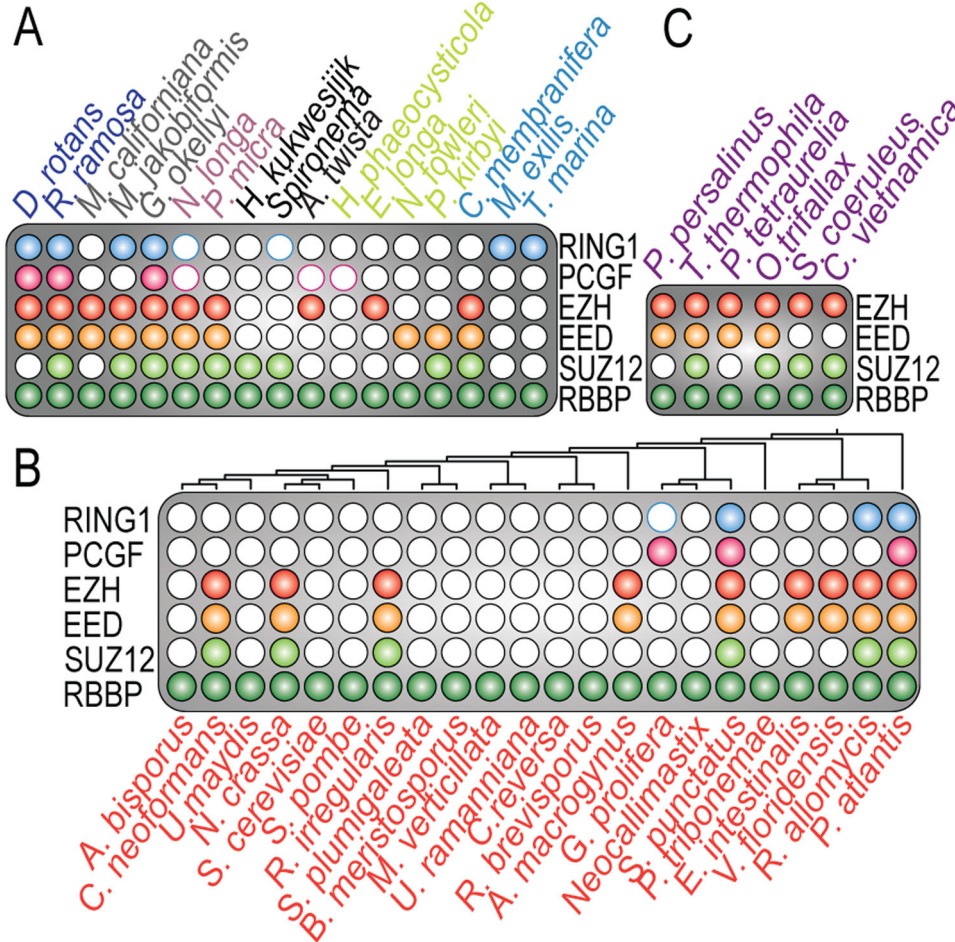

**Fig. 3 Phylogenetic profiles of the core subunits of PRC1 and PRC2. A** Phylogenetic profiles of species in major eukaryotic groups proximal to the eukaryotic root, solidifying their presence in LECA. **B** Representation of the presence of PRC1 and PRC2 core subunits in fungi, a group that was previously believed to have lost PRC1 completely. **C** Phylogenetic profile of ciliates, highlighting the detection of highly diverged EED sequences.

Metamonada, Malawimonadida, CRuMs, Hemimastigophora, and Provora (Fig. 2, S1A, B).

Although the eukaryotic group of fungi is generally believed to have lost PRC1[2], we uncovered PRC1 orthologs in multiple deep branching fungal species. For instance, we observed PRC1 (either RING1 or PCGF) in the deep branching fungi *Rozella allomycis* and *Gonapodya prolifera* (Figs. 2, 3B). Moreover, we confirmed the presence of PRC1 orthologs in the fungus *Spizellomyces punctatus*[64]. Collectively, our phylogenetic analysis reveals that the PRC1 core was already present in LECA and remained highly conserved throughout eukaryotic life (Fig. 2, SI Fig. S2).

We additionally identified that canonical PCGF2/4 and non-canonical PCGF1, 3, 5, and 6 form monophyletic sister groups due to a duplication event at the root of metazoan evolution, which is in line with a previous study (Fig. S1B)[52]. However, our findings also propose the possibility of subsequent duplications occurring within PCGF1, 3, 5, and 6 at the root of metazoan evolution. This inference arises from the observation that various homologs within this cluster, notably from *Nematostella vectensis*, *Branciostoma belcheri*, and *Crassostrea gigas*, cluster distantly.

Addressing the question of whether the ancestral PCGF leaned more towards the canonical or non-canonical type, our results indicate repeated annotations of PCGF in the sister group of animals (Choanoflagellates) and other early branching species, with these annotations predominantly corresponding to PCGF5 or PCGF3 according to eggNOG. This implies a possibly greater sequence similarity with PCGF5 compared to other PCGF paralogs.

However, we approach these findings cautiously, refraining from definitive conclusions about the presence of a specific ancestral canonical or non-canonical PCGF variant. Instead, we suggest that the ancestral PCGF should not be conclusively characterized as canonical or non-canonical in nature, and believe that the diversification between non-canonical and canonical PCGF was a later development in evolution.

**Evidence for the presence of PRC2 in LECA**. To reconstruct the evolution of PRC2 we traced the functional core of PRC2, consisting of EZH, EED, SUZ12 and RBBP[12,13,65], across our diverse set of eukaryotes. Similar to PRC1, we studied the domain conservation of these subunits. Because Pfam domain profiles (v.35) again failed to detect key domains in a substantial number of orthologs, we curated our HMM profiles by combining sequence alignments with AlphaFold2 structural predictions (Fig. 1B, SI Data and Methods and Supplementary Data 2). Within our selection of 178 eukaryotes, we detected 94 species with orthologs for EZH, 91 with EED, 87 with SUZ12, and 178 with RBBP (Fig. 2, SI Fig. S1C–E). In accordance with our phylogenetic analysis of PRC1, the phylogenetic trees of PRC2 subunits also exhibited numerous orthologous groups with extensive species overlap. This observation thus similarly enables the reliable inference of the speciation node and the presence in LECA for the PRC2 subunits. EZH serves as the key enzymatic subunit and is a crucial indicator of the presence of a functional PRC2[66]. In our

phylogenetic profile, EZH, EED, and SUZ12 exhibited strong co-occurrence. However, the co-occurrence of RBBP with other PRC2 was less prominent, owing to its involvement in various other protein assemblies[67]. The phylogenetic trees of EZH and RBBP had the highest support values throughout all major eukaryotic groups (SI Fig. S1C–E). In contrast, the phylogenomic delineation of EED orthologs was more ambiguous, due to their low sequence conservation which results in trees with lower resolution as well as difficulties in finding homologs and annotating the tree with protein domains (SI Fig. S1D). For instance, PRC2 was recently experimentally identified in the ciliates *Paramecium tetraulia* and *Tetrahymena thermophila*[56,57], but the EED sequences in these organisms were undetectable by standard sequence-based homology methods. By performing a HMM search with a query based on the profile of the sequence alignment of these two sequences however, we retrieved EED orthologs in most ciliates, but not in the Alveolates *Stentor coeruleus* and *Colponemida vietnamica* (Fig. 3C).

Like EED, SUZ12 was also found to be highly divergent in various species, e.g., a recent study demonstrated that MES-3 in *Caenorhabditis elegans* is a diverged SUZ12 ortholog rather than a *Caenorhabditis elegans* specific addition to PRC2[68]. The N-terminus of SUZ12 in animals contains five motifs: a zinc-finger binding (ZnB), WD-domain binding 1 (WDB1), C2 domain, zinc finger (Zn), and WD-domain binding 2 (WDB2) (Fig. 1B)[12,66,69]. All these motifs are highly conserved throughout animals and plants. Pertinently, the C-terminal VEFS box of SUZ12 is crucial for the formation of the PRC2 catalytic lobe[12,66,69]. Therefore, we used the VEFS-box as the basis for our inference analysis to systematically detect SUZ12 orthologs. This approach uncovered a strong correlation between SUZ12 orthologs and the phylogenetic profiles of the other PRC2 subunits (SI Fig. S2). Note that examples of SUZ12-like orthologs without a VEFS-box were recently described, e.g., *Paramecium tetraulia* has an adapted SUZ12-like ortholog without a VEFS-box[57]. Consequently, we could not detect these SUZ12-like orthologs in our phylogenetic inference (Fig. 3C). In addition, a previous study reported that in the yeast *Cryptococcus neoformans*, SUZ12 is substituted by a structurally unrelated protein, Bnd1[70]. Indeed, we initially missed SUZ12 ortholog in the yeast *Cryptococcus neoformans*. However, by combining HHpred searches with AlphaFold2 structural predictions, we found that, like MES-3 in *Caenorhabditis elegans*, Bnd1 is most likely a highly diverged SUZ12 ortholog (SI Fig. S3). Further inferred absences of subunits predicted by our analysis might thus have been substituted by other, currently unknown, subunits, or were not detected due to high sequence divergence (e.g., EED, SUZ12). Consistent with the notion that EZH, EED and SUZ12 comprise the crucial functional core of PRC2, these subunits typically co-occur across the eukaryotic tree of life, reflecting the close and ancient biochemical interdependence of EZH, EED and SUZ12.

**Intra-complex evolution of PRC1 and PRC2 is cohesive while inter-complex evolution is uncoupled.** The evolution of RING1 and PCGF is largely coupled, i.e. these core PRC1 subunits were mostly lost or retained together during eukaryotic evolution (Fig. 4A). Likewise, the genes encoding the core of PRC2 were typically lost or retained together. This suggests that both PRC1 and PRC2 function and evolve as cohesive units. To obtain a quantitative measure of their cohesiveness throughout eukaryotic evolution, we calculated the average Pearson's correlation coefficient between the phylogenetic profiles of the subunits of PRC1 and PRC2 (Fig. 4B). We found that the subunits of the PRC2 core (EZH, EED, and SUZ12) had a high average correlation ($r = 0.73$), as did the core subunits of PRC1 (RING1 and PCGF, $r = 0.7$) (SI Fig. S4). This confirms that these subunits tend to evolve together

within their resident complexes. As expected, given its participation in multiple additional chromatin regulating pathways, RBBP only correlated weakly with other PRC2 subunits.

These observations raise the question whether the evolution of PRC1 and PRC2 might be coupled too. To address this question, we scored the presence of PRC1 or PRC2 only if all defining core subunits were present (i.e., both RING1 and PCGF for PRC1, and EZH, EED and SUZ12 for PRC2). Our analysis of 178 eukaryotes, selected to cover all known major eukaryotic groups, revealed that 40 species retained both PRC1 and PRC2, 12 lost PRC2 but retained PRC1, 17 retained only PRC2, whereas 52 lost both complexes. The Polycomb status of the remaining 57 eukaryotes remains unclear because they contained sequences without a recognizable RAWUL domain or harbored apparently partial PRCs. Thus, ~33% of species we analyzed contains remnants of the Polycomb system but lacks an unambiguous complete PRC1 or PRC2. About 22% of eukaryotes retained both PRC1 and PRC2, ~16% of eukaryotes retained either PRC1 or PRC2, without a clear preference for one over the other, while ~29% lost both. Thus, a substantial portion (~42%) of organisms that contain (part of) the Polycomb system use only PRC1 or only PRC2. We expect that with increasingly sensitive detection, the degree of uncoupling between PRC1 and PRC2 might turn-out to be even higher than presented in this study. The presence of either only PRC1 or only PRC2 was particularly noticeable in Amoebozoans, fungi, Cryptista, Haptista, Rhodophyta, Glaucophyta, Chloroplastida, and SAR, but we were unable to determine a consistent trend of either complex being more likely to be lost or retained (Fig. 4A). In conclusion, while PRC1 and PRC2 were both present in LECA, their evolution throughout eukaryotic life has been largely uncoupled. Concurrent conservation of both complexes was most frequent in land plants and animals, while other major taxa displayed a more diverse pattern of conservation for PRC1 and PRC2 (Fig. 4A). In contrast to their high intra-complex correlations, PRC1 and PRC2 subunits only displayed a low degree of inter-PRC correlation ($r = 0.4$). Collectively, these results reflect great evolutionary flexibility of the Polycomb system.

The evolutionary flexibility of the system is similarly reflected in the large number of species that lack PRC1 or PRC2. While the possibility exists that instances where the detection of these complexes was missed could be associated with incomplete or inaccurate genome assembly, the loss of a crucial system such as Polycomb aligns with observations of loss in other crucial molecular mechanisms across different organisms. Notable instances include the ska system, minor spliceosome, and cilia, which all have an uneven distribution among species due to the influence of gene loss pressures throughout genome evolution[71–74]. In fact, recent research has also brought to light that in Plasmodium, the experimentally verified absence of PRCs has led cohesin to perform the role of gene regulation, highlighting the potential of alternative mechanisms substituting the Polycomb system[75].

**Conservation of residues involved in H2AK199ub - but absence of H2AK199ub marks – suggests PRC2-independent functions of PRC1.** Studies in human systems have revealed that H2AK119ub by PRC1 is particularly dependent on several key residues in the catalytic subunit RING1. Specifically, HsRING1B residues I53 and D56 are essential for stabilizing the interaction with the E2 ubiquitin ligase UbcH5c[76], and K97 and R98 mediate the interaction with nucleosomes[77]. Both are essential for H2AK119ub in humans[22]. Of these key residues, I53, K97, and R98 are particularly well-conserved across RING1 orthologs in eukaryotes (SI Fig. S5), implying that the catalytic function of PRC1 could be conserved across species.

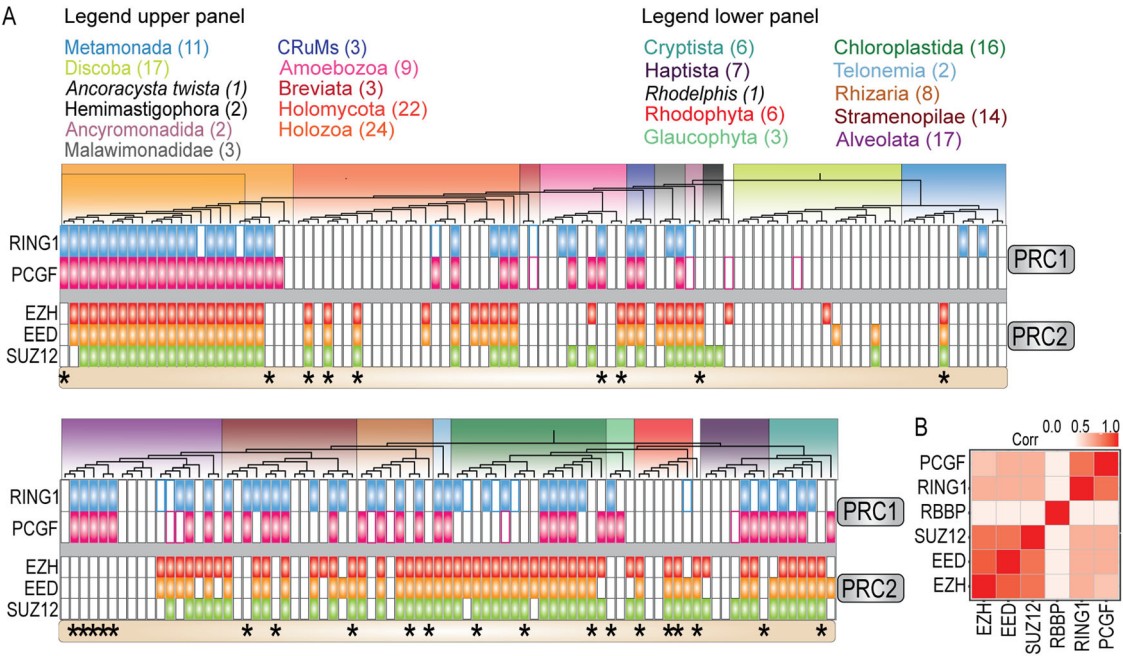

**Fig. 4 Intra-complex evolution of PRC1 and PRC2 is cohesive, while inter-complex evolution is largely uncoupled. A** Phylogenetic profiles of PRC1 and PRC2 core subunits in all independent species included in our analysis. The profiles show the overall cohesive loss- or retention of intra-complex subunits, and the largely uncoupled evolution of PRC1 and PRC2. RBBP was not included because it does not represent the presence or absence of the PRC2. Data was split into two panels and subunits are colored as follows: RING (blue), PCGF (Pink), EZH (Red), Orange (EED), SUZ12 (Green). Asterisks highlight species where either PRC1 or PRC2 is uncoupled. **B** Heatmap of the correlations between the subunits. Intra-complex subunits have a higher correlation than inter-complex subunits. Full names of all species are provided in SI Fig. S2.

By contrast, D56 is only conserved in metazoan orthologs, suggesting that if PRC1 indeed has catalytic functions outside of metazoans, it is not dependent on D56 in non-metazoan taxa. Notably, RING1 orthologs in the plant *Arabidopsis thaliana* have a glycine at this position in the RING motif, and previous experimental works have shown that these orthologs do mono-ubiquitinate H2A in this species[78]. Therefore, it appears that *Arabidopsis thaliana* RING1 is capable of ubiquitinating H2AK119 independent of the role D56 plays in metazoans. More broadly, the equivalent histone post-translational modification (hPTM) of H2AK119ub has so far not been experimentally detected outside of holozoa and *Arabidopsis thaliana*[64]. Thus, it remains unclear whether the conservation of I53, K97, and R98 across the breadth of eukaryotic diversity indeed correlates with widespread catalytic capacities of RING1 orthologs, or if those functions are truly limited to H2AK119ub in holozoa and *Arabidopsis thaliana*. This raises intriguing possibilities for the PRC1 complex to play PRC2-independent roles in gene regulation and chromatin modification in a wide range of organisms, beyond the traditionally associated functions in metazoans.

**Structural similarity between RYBP and CBX suggests a common ancestor**. Both PRC1 and PRC2 form subcomplexes through the incorporation of accessory subunits or through the combinatorial assembly of paralogous components[4,6–8]. ncPRC1 and cPRC1 are distinguished by the mutually exclusive binding of either CBX (cPRC1) or RYBP (ncPRC1)[20]. CBX harbors a chromodomain at its N-terminus and a CBX7_C domain at its C-terminus, while RYBP contains a zf-RanBP domain at its N-terminus and a YAF2 domain at its C-terminus (Fig. 5A). Interestingly, both the CBX7_C- and YAF2 domain form a β-hairpin of around 16 amino acids. From an evolutionary perspective, mutual exclusivity (such as between RYBP and CBX) is often found between paralogs that define variant complexes[79,80].

Therefore, we sought to investigate if CBX and RYBP might actually be distant paralogs by determining whether there is structural or highly diverged sequence homology between them. Whereas we did not detect full length structural- or sequence homology between CBX and RYBP, a structural alignment of their C-terminal β-hairpins using the TM-align tool[81] revealed that they superimpose almost perfectly (RMSD = 2.35 Å) (Fig. 5A, Supplementary Data 3). Moreover, a sensitive sequence profile-vs-profile (HHpred)[82] search with the C-terminal motifs of RYBP and CBX8 revealed that they were reciprocal best hits (Fig. 5B, Supplementary Data 3). Based on this evidence, we propose that the domains CBX7_C and YAF might be distant paralogs. Thus, to avoid any confusion resulting from the use of multiple names referring to the same domain, we will use β-HP throughout the remainder of this article. Combined, the structural similarity and the reciprocal best hits of their β-hairpin domains are first indications that CBX and RYBP might be switching paralogs in PRC1.

**Evolutionary evidence for the occurrence of ncPRC1 prior to cPRC1**. To further study the emergence of ncPRC1 and cPRC1, we investigated the evolutionary origin of their signature subunits RYBP, CBX, and PHC. Recently, a putative ortholog of RYBP, containing only the N-terminal zf-RanBP motif, was identified in the Choanoflagellate *Salpingoeca rosetta*, which is a close unicellular relative of animals[52]. Additionally, we identified full length orthologs of RYBP in two Choanoflagellata, *Salpingoeca kvevrii* and *Diaphanoeca grandis*. These orthologs contain both the N-terminal zf-RanBP motif as well as the C-terminal β-hairpin, suggesting that these are bona fide RYBP orthologs (Fig. 6A, Supplementary Data 4, 5). We also detected a putative ortholog containing only the N-terminal zf-RanBP motif in the nucleariid *Parvularia atlantis*, which is a close relative of fungi (Fig. 6A). In contrast, we were not able to detect orthologs of CBX outside of animals (Supplementary Data 4).

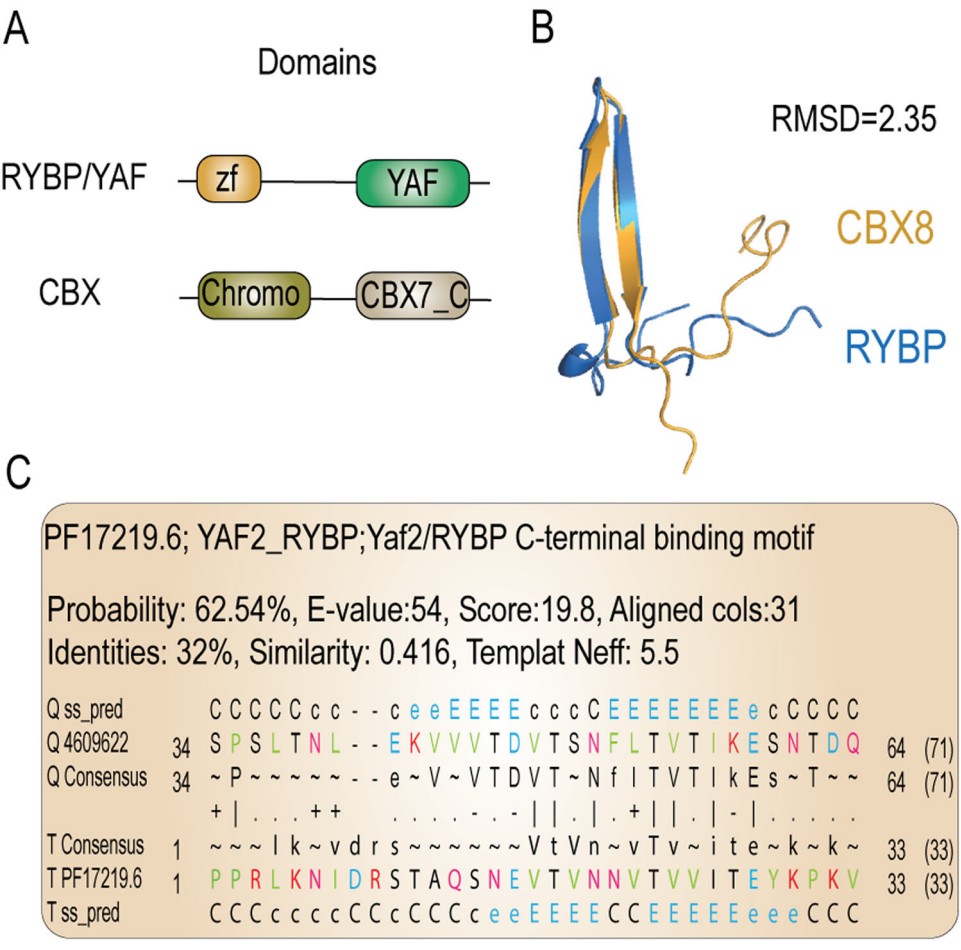

**Fig. 5 Structural homology and reciprocal best hits suggest that the β-hairpin of RYBP and CBX are orthologous. A** Schematic representation of RYBP/ YAF and CBX. Both the C-terminal YAF domain and CBX7_C domain consist of a β-hairpin. **B** Structural alignment of the β-hairpin of RYBP, and CBX. The structures superimpose almost perfectly (RMSD = 2.35). **C** Output of the YAF2 C-terminal β-hairpin as the best hit of a sensitive sequence profile-vs-profile (HHpred) search with the β-hairpin of CBX.

Thus, it appears that RYBP originated before CBX. Next, we performed a phylogenetic analysis of SAM domains which are present in the PRC1-associated proteins PHC, SCML, SCMH, L3MBTL, SFMBT and MBTD (SI Data and Methods Supplementary Data 4). The SAM domains clustered together in a monophyletic group, reflecting a shared ancestry (SI Fig. S1F), but could be separated further into two distinctive groups; one that consisted mostly of proteins containing an MBT domain, and one without such a domain (Fig. 6B). We therefore next performed a phylogenetic analysis of MBT domains with the aim to find orthologs outside of animals that would imply an earlier origin for these proteins (SI Data and Methods and Supplementary Data 4). Our analysis revealed that the MBT domains of the PRC1 accessory subunits were part of an orthologous group. Strikingly, we identified two sequences in the relatives of animals and fungi that contained both a potential SAM- and a MBT domain: one filasterean, *Capsaspora owczarzaki*, and the nucleariid *Parvularia atlantis* (SI Fig. S1G). Although these sequences most likely encode orthologs of the L3MBTL/SCML PRC1 accessory subunits, the SAM- and MBT domains are in reverse order compared to their animal counterparts (Fig. 7). Adding the *Capsaspora owczarzaki*, and *Parvularia atlantis* SAM domains to our earlier inference analysis did not yield a monophyletic clustering with the SAM domains of SCML, SCMH, L3MBTL, and PHC (SI Fig. S1F). This negative result might be due the limited phylogenetic signal of the relatively small SAM

domain (65–70 amino acids). Thus, the orthology of the filasterean, *Capsaspora owczarzaki*, and the nucleariid *Parvularia atlantis* SAM domains remains unclear.

In summary, our collective results suggest that the ancestor of animals and fungi might have already harbored a RYBP and L3MBTL-associated ortholog (Fig. 7). We propose a scenario in which RYBP was lost in fungi and Filasterea, but retained in Nucleariidae, Choanoflagellata and animals. Furthermore, we hypothesize that CBX arose via a gene duplication of RYBP at the root of animals, after which the zf-RanBP was replaced by a chromo-domain (Fig. 7). Similarly, our data suggests that the SAM domain containing accessory subunits of PRC1 most likely originated from a singular gene rearrangement event of an ancestral L3MBTL-associated ortholog. The occurrence of RYBP and L3MBTL-associated proteins in these early branching lineages suggests that ncPRC1 defining subunits, as we know them in animals, originated before cPRC1 accessory subunits.

## Discussion

Here, we present a systematic characterization of the Polycomb system throughout the eukaryotic tree of life. Phylogenetic and structural analysis of 178 eukaryotes representing all known major eukaryotic groups revealed broad conservation of both PRC1 and PRC2. Our results imply that PRC1 and PRC2 were both present in LECA, but that their subsequent evolution is uncoupled. Sensitive

analyses allowed us to identify highly diverged orthologs, which previously escaped detection. Contrary to the general believe that fungi have lost PRC1, we uncovered PRC1 orthologs in multiple deep branching fungal species. Furthermore, we identified orthologs that had previously remained undetected because of their significant sequence divergence, such as EED in ciliates and SUZ12 orthologs in *Cryptococcus neoformans* and *Caenorhabditis*

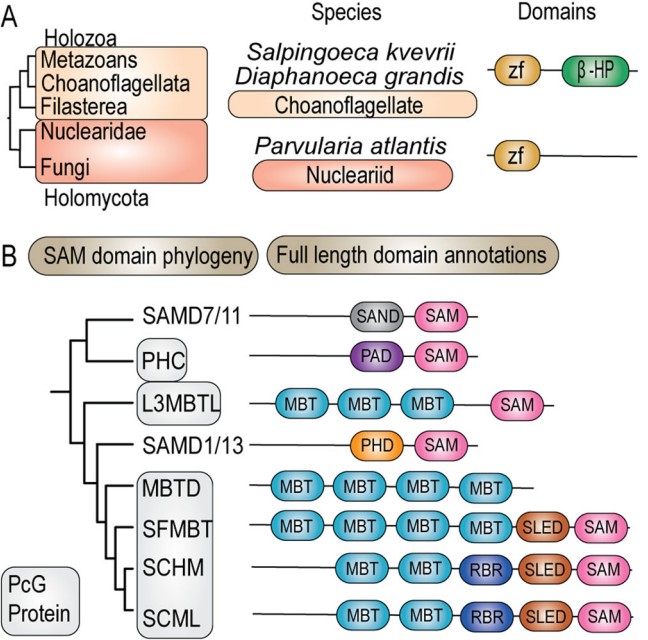

**Fig. 6 Phylogenetic history and domain organization of RYBP and SAM-domain proteins. A** Domain organizations of the identified RYBP orthologs in the relatives of animals and fungi. **B** Schematic representation of the two clusters in the SAM domain phylogeny (SI Fig. S1F) and the full-length domain annotations of the identified proteins.

*elegans*[68]. We expect that increasingly sensitive techniques will uncover additional species harboring divergent Polycomb orthologs or other, currently unknown, subunits. Thus, the evolutionary conservation, uncoupling, and flexible development of the Polycomb system might be even more prevalent than presented here. Incomplete or inaccurate genome assemblies may have also falsely suggested the absence of certain Polycomb complexes, and led to an over-estimation of uncoupled evolution. However, we have made extensive efforts to enhance the quality of our eukaryotic database, thereby minimizing the chance that we falsely infer absences (SI Data and Methods). Finally, we identified orthologs of ncPRC1-signature domains, but not cPRC1-specific domains, in unicellular relatives of animals and fungi. Based on the distant homology between the C-terminal domains of RYBP and CBX, we propose that these subunits might be switching paralogs. Moreover, we propose that the origin of ncPRC1 predates that of cPRC1, adding weight to earlier hypotheses that ncPRC1 accessory subunits originated before cPRC1[2].

Ever since their discovery, the relationship between PRC1 and PRC2 has been debated. The binding of CBX subunits of cPRC1 to H3K27me3, and the recognition of H2AK119ub1 by PRC2.2-associated factors JARID2 or AEBP2, has inspired multiple hierarchical or interdependent recruitment models[2,4,6–9,27]. However, in spite of cooperative cross-talk, in vivo studies showed that there is no crucial dependency between PRC1 and PRC2 that is essential for repression of canonical Polycomb targets[31–35,38]. Rather, PRC1 and PRC2 function in largely redundant pathways, thereby ensuring robust gene silencing. Our analysis showed that a substantial portion (~42%) of organisms that contain (part of) the Polycomb system use only PRC1 or only PRC2. These results provide compelling evolutionary support for the predominantly independent function of PRC1 and PRC2.

While the central function of H3K27me3 for Polycomb repression in animals is unequivocal[32,36], the role of H2Aub1 turned-out to be more enigmatic. During mouse and *Drosophila* development, H2Aub1 contributes, but is not absolutely required for the silencing of Polycomb targets[32,38–40]. Rather, it appears that higher order chromatin organization by cPRC1, which is independent of

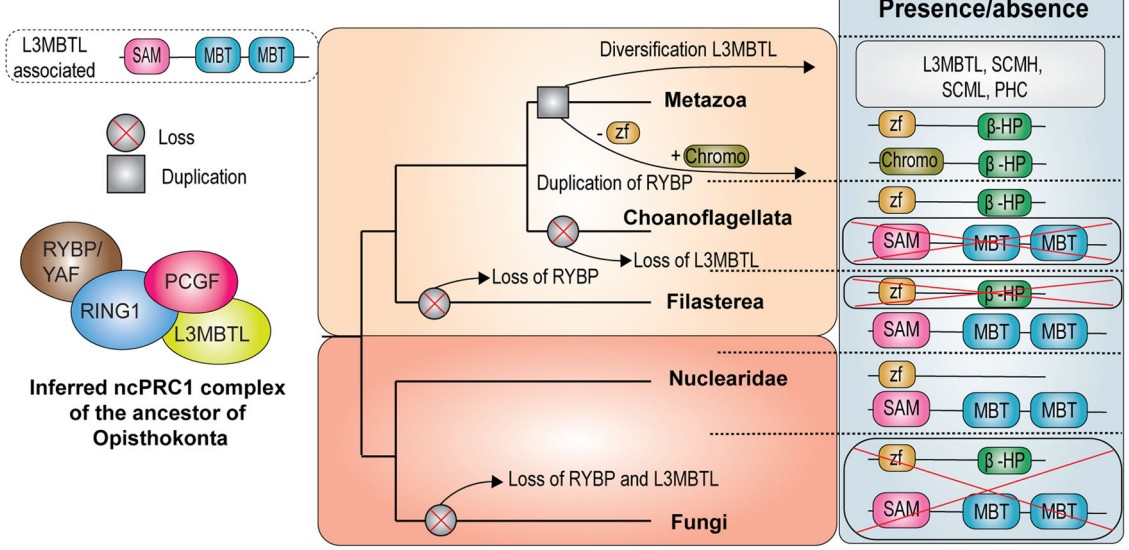

**Fig. 7 ncPRC1 evolved prior to cPRC1.** The ancestor of animals and fungi most likely already had a distinct ncPRC1 functionality, while cPRC1 originated later in the root of animals. The upper left panel presents a schematic overview of the identified L3MBTL-associated ortholog. Shown is furthermore an evolutionary scenario of the origin and evolution of ncPRC1 and cPRC1 accessory subunits. Our scenario suggests that RYBP was lost in fungi and Filasterea, and retained in Nucleariidae, Choanoflagellata and animals, while CBX arose via a gene duplication of RYBP at the root of animals, after which the zf-RanBP got replaced by a chromo-domain. SAM domain containing accessory subunits of PRC1 likely originated from diversification of the ancestral L3MBTL-associated ortholog, after duplications.

H2Aub1, is crucial for Polycomb function in animals[32,38–40]. Supporting this notion, phylogenetic profiling of histone modifications revealed broad conservation of H3K27me3, while H2Aub1 is mainly restricted to animals and plants[64]. Thus, while PRC1 is highly conserved and can be traced back to LECA, H2Aub1 is not. This implies that the primary function of PRC1 does not depend on H2Aub1. Collectively, these observations argue that biochemical cross-talk between PRC1 and PRC2 is a secondary development during the evolution of animals.

In agreement with our phylogenetic analysis, functional studies in organisms other than animals revealed that the Polycomb system is highly plastic and employs diverse mechanisms of action. The Polycomb system in plants is the result of an independent evolutionary trajectory, yielding unique PRCs, which are absent in animals. e.g., plants lack cPRC1 but contain a wide variety of plant-specific PRC1 and PRC2 accessory subunits. One of the plant-specific PRC2-associated proteins, like-heterochromatin protein 1 (LHP1) contains a chromo domain and chromo-shadow domain that bind to H3K27me3[83,84]. Thus, LHP1 might mediate a positive feedback loop that stimulates the formation of H3K27me3 domains[4,7–9]. The EZH ortholog in ciliates, Ezl1, is a dual-specificity methyltrasferase that mediates both H3K27me3 and H3K9me3[85]. Ezl1 binds a PRC2 core associated with the RNA interference effector Ptiwi09, and RING-finger proteins Rnf1 and Rnf2[56,57,86]. Thus, the Ezl1 complex appears to resemble a combination of PRC2 and PRC1. However, Rnf1 and Rnf2 are not orthologous to the canonical RING1 proteins found in PRC1 and are not required for H2Aub1, but they do contribute to histone methylation by the Ezl1 complex. Ptiwi09 mediates small-RNA-guided recruitment of the Ezl1-PRC2 complex to transposons, where it deposits H3K27me3 and H3K9me3, triggering transposon elimination[56,57]. Examples of adaptation in the absence of PRC1 (Fig. 3B) are provided by the filamentous fungus *Neurospora crassa* and the yeast *Cryptococcus neoformans*. In *Neurospora crassa*, the BAH- and PHD-domain protein EPR-1 acts as a H3K27me2/3 reader. EPR1 is a crucial protein involved in H3K27me3-directed silencing and mediates the formation of nuclear foci resembling Polycomb bodies[87]. The *Cryptococcus neoformans* PRC2-associated chromo-domain protein Ccc1 recognizes H3K27me3, which is crucial for its localization to repressive subtelomeric domains[70]. Thus, in the absence of PRC1, other proteins recognize H3K27me3 to effectuate transcriptional silencing. Finally, it should be noted that some Polycomb factors have been repurposed for additional, chromatin-independent functions. E.g., *Drosophila* PSC uses a diptera-specific domain to bind and regulate cyclin B, thereby controlling cell cycle progression (SI Fig. S6)[88]. Mammalian SCML2B regulates the G1/S checkpoint by binding and modulating the CDK/Cyclin/p21 complex[89]. These examples from diverse eukaryotes emphasize the remarkable evolutionary flexibility of the Polycomb system.

What was the original function of the Polycomb system? There is accumulating evidence that the primary purpose of PRC2 in unicellular eukaryotes is the silencing of transposable elements to protect genome integrity[56,90–93]. Similarly, in addition to gene control, plant PcG proteins ensure transcriptional repression of repetitive genomic elements and transposable elements[90,91,94–98]. Collectively, these observations give rise to the hypothesis that gene-selective transcriptional repression by the Polycomb system evolved from an initial function in silencing of transposal elements[58]. Additional experimental studies will be necessary to establish the ancestral functions of PRC1 and PRC2, and delineate how these evolved towards their role in gene regulation in animals and plants. The findings presented here provide unique insights into the evolutionary development of the Polycomb system and its function in genome regulation. Additionally, it provides a framework for experimental analysis of the

Polycomb system in diverse eukaryotic organisms. We anticipate that these future studies will shed light on the connection between genome evolution and functional adaptation of PRC1 and PRC2.

## Materials and methods

**Proteome database**. To study the occurrences of PRC1 and PRC2 genes across the eukaryotic tree of life, a reference dataset of predicted eukaryote proteomes comprising 178 diverse eukaryotic species was assembled. A detailed description of the included species and how the data was collected is described in SI Data and Methods and Supplementary Data 1. Briefly, species were selected based on multiple criteria; i) we attempted to include representatives of all major eukaryotic groups encompassing most currently known eukaryotic diversity; ii) we weighed the species against genome quality statistics (BUSCO); if multiple proteomes or different representatives of a single species were available, the most complete one was selected; and iii), in clades with many closely related, high quality genome assemblies, we selected important (model) organisms.

To avoid incomplete or improper genome assembly, BUSCO (v5.2.2; eukarya_odb10) was used to determine the quality of the proteomes of our selected species. Single-copy BUSCO orthologs found in at least 75% of all species were selected and taken as marker genes for the construction of a phylogenetic tree, yielding a comprehensive resolution of the diversity in this combined set. The phylogenetic tree was inferred using IQTree (v2.2.0) using the LG+F+R15 substitution model. The resulting phylogeny was visualised in iTOL, and BUSCO scores were plotted on these allowing for manual selection of species at key phylogenetic positions with the highest available quality proteomes across the relevant taxa[99].

**Homolog detection and phylogenetic inferences of core subunits PRC1 and PRC2**. To study the evolutionary history of the PRC1 and PRC2 subunits, a general homolog detection and phylogenetic inference method was initially applied and iterated upon if needed. Briefly, eukaryotic homologs were identified using pairwise searches with BLASTP v.2.12.0+[100] against our local proteome database with full length *Homo sapiens* sequences as queries (Supplementary Data 6). At most 500 significant hits (default: 1e-5) were aligned using MAFFT v.7.490 (settings: genafpair, maxiterate = 1000) and processed with trimalAl v1.4.rev15 [gt = 0.1][101,102] (Supplementary Data 7). Phylogenetic trees were inferred with IQ-TREE v.2.1.4 (ModelFinder, ultrafast bootstrap = 1000)[103], and subsequently visualized and annotated using iTOL (SI Fig. S1A–G, Supplementary Data 7)[99]. To manually delineate orthologous groups from the phylogenetic trees, all sequences in the trees were annotated with independent function predictions (eggnog) and predicted domains based on HMM searches with the 'hmmsearch' tool from the HMMER package (http://hmmer.org/, HMMER 3.3.2), with profiles from Pfam (v.35)[60,104]. Delineation of the orthologous group was based on i) support values (>90); ii) consistency of domain annotations; and iii) consistency of independent function predictions. If all three sufficed, all sequences in the group were considered to constitute an orthologous group.

In some cases, interpretation of the phylogenetic trees was hampered by highly diverged sequences, incomplete genomes, low support values, or low sensitivity of Pfam profiles (v.35) that were unable to detect protein domains. Subsequent manual inspection of homologues by AlphaFold2 structures through ColabFold and multiple sequence alignments revealed the likely presence of known, yet highly diverged, proteins domains[61–63]. In these cases, custom profiles for domains were created with the 'hmmbuild' command from the HMMER package (http://hmmer.org/, HMMER 3.3.2) (Supplementary Data 2). These custom profiles were used to

annotate sequences included in the respective phylogenetic trees, which enabled the detection of proteins domains that were previously reported to be absent, and therefore supported our orthologous inference. For some analyses, we also performed HMM searches with full length profiles. The specific analyses for these subunits, and supporting data, are described in SI Data and Methods and Supplementary Data 4. Fasta sequence files of our orthologous groups are provided in Supplementary Data 8. AlphaFold2 predicted structures used in this study are provided in Supplementary Data 5.

**Structural analysis of CBX and RYBP.** To detect structural similarity between the C-terminal regions of CBX and RYBP, the pdb_selres.py script from pdb-tools (https://wenmr.science.uu.nl/pdbtools/) was used to cut out the β-sheet motifs. Accordingly, we used TMalign tool (https://zhanggroup.org/TM-align/) to perform a structural alignment[81] (Supplementary Data 3). Additionally, a single blast was performed with the β-sheet motifs using the MPI bioinformatics toolkit (https://toolkit.tuebingen.mpg.de/) and the MSA was used as a query to perform a HH search (https://toolkit.tuebingen.mpg.de/) against the Euk_Homo_sapiens_04_Jul_2017 proteome, with PDB_mmCIF70_12_Aug, Pfam-A_v35, and SMART_v6.0 as structural/domain databases[82,105,106] (Supplementary Data 3).

**Correlation plot quantification and visualization.** To study whether PRC1 and PRC2 co-evolved, a correlation plot of the phylogenetic profiles was created. For this correlation plot, putative orthologs were excluded, i.e. they did not contribute to the presences nor the absences. Phylogenetic profiles were created by setting a '1' for presence and '0' for absence. To then evaluate the co-occurrence of PRC1 and PRC2, the similarity of their phylogenetic profiles was quantified by using the Pearson correlation coefficient which for binary data is equivalent to the phi coefficient. Genes were clustered based on their phylogenetic profiles using average linkage and the resulting heatmap was visualized with ggplot in R version 4.1.2[107].

**Statistics and reproducibility.** All details regarding the reproducibility and statistics of our analyses are provided in the main text and in the 'Materials and Methods' section and in SI Data and Methods and Supplementary Data. For BLASTP searches we used the default of 1e-5 as significant cut-off. To evaluate the co-occurrence of PRC1 and PRC2 we used the Pearson correlation coefficient which for binary data is equivalent to the phi coefficient.

**Reporting summary.** Further information on research design is available in the Nature Portfolio Reporting Summary linked to this article.

## Data availability

The data supporting the conclusions made in this article are available via the Supporting Information and referenced Supplementary Datasets. Numerical source data of reliable orthologs for Fig. 2 is provided in Supplementary Data 9. Data of putative orthologs was manually added to the pie charts based on data in SI Fig. S2.

## Code availability

Scripts to remove splice variants from the proteome database are available upon reasonable request to the corresponding author.

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

## Acknowledgements
We thank Eelco Tromer, Jolien van Hooff and Carlos Santana-Molina for advice in compiling the eukaryotic proteome database. We thank Carlos Santana-Molina for sharing scripts to remove splice variants from the proteome database. This work was supported by the Netherlands Organisation for Scientific Research (NWO-Vici 016.160.638, to B.S.).

## Author contributions
Author contributions: B.d.P., M.F.S., P.V., and B.S. designed research; Bd.P. performed research; Bd.P., analyzed data; M.W.D.R., developed the methods for, and created, the proteome database; and B.d.P, M.W.D.R., M.F.S., P.V., and B.S. wrote the paper. All authors approved the final version of the manuscript.

## Competing interests
The authors declare no competing interests.
