## [Peer Review File · Communications Biology]

Referee expertise:

Referee #1:genomics expert

Referee #2:polycomp epigenetics expert

Referee #3:evolution of histone variants

Reviewers' comments:

Reviewer #1 (Remarks to the Author):

Here the authors present a comprehensive analysis of the evolution of the Polycomb group proteins from the Last Eukaryotic Common Ancestor throughout the eukaryotic tree, by looking for the presence of the main PRC1 and PRC2 core components in diverse eukaryotes. Their approach (including manual curation of the domains eg RING, RAUWL, VEFS , to avoid the lack of sensitivity from default Pfam profiles) has led to the identification of high-confidence orthologs including ones missed in previous studies and in organisms thought to have lost orthologs. The authors show that over evolution the core components within a PRC complex are either lost or retained together, but between the complexes are uncoupled, with many organisms only having either PRC1 or PRC2. The authors also show that RYBP (in ncPRC1) appears to have originated before the structurally similar CBX protein (in cPRC1). Taken together, this suggests an independent function of PRC1 and PRC2 and that the crosstalk between complexes, especially CBX binding to K27me3, is a secondary development in evolution.

The work presented here is a very thorough search for presence of PRC1 and PRC2 core components throughout evolution of eukaryotes. Using manual curation by searching alignments of the known protein domains, rather than just relying on Pfam profiles, has shown the presence of PRC1 and PRC2 (and indeed loss in some lineages) in more organisms than previously thought which is very valuable and clears up the literature. It is especially interesting when thinking about the mechanism of repression of these complexes and suggests that the two protein complexes can work independently. In terms of the models of recruitment to chromatin, the finding that the ncPRC1 complex evolved before cPRC1, adds weight to the argument that PRC1 can work independently of PRC2 and can be recruited without the need of CBX proteins to bind to the H3K27me3 of PRC2. This is an important observation.

In humans, there are 6 PCGF proteins, and indeed the authors look at the phylogenetic tree of PCGF protein in FigS1b, although when each of these different proteins evolved wasn't discussed in the text. Which type of human PCGF protein did the ancestral one look most like? This would be an interesting point to expand on, particularly because the type of PCGF protein can also determine if the PRC1 complex is canonical or non-canonical.

The authors also only looked at the core components of PRC2 but it would be interesting to either look for the presence of the accessory proteins eg Jarid2, Aebp2 or PCL proteins, or in the least add a comment to the text about the evolution of the different types of PRC2 complexes (in a similar way to ncPRC1 and cPRC1).

Reviewer #2 (Remarks to the Author):

The manuscript by de Potter et al. is a study on the phylogenetic of Polycomb proteins. Exploiting the growing collection of eukaryotic genomes the authors search throughout 178 species for possible

evolutionary ancestors of core subunits of the two major classes of Polycomb Repressive Complexes, PRC1 and PRC2. It extends significantly previous observations on Polycomb phylogeny resulting from more detailed work adding new information of interest to researchers in the field and, in particular, to those attracted by molecular evolution.

PRC core complexes make enzymatic modules responsible for histone modifications: a heterodimeric ubiquitin E3 ligase for lysine 119 of histone H2A in PRC1, and the SET domain lysine methyltransferase for lysine 27 of histone H3 (H3K27me3), plus accessory subunits, in PRC2. Searching for combinations of protein motifs found in core subunits results in meaningful, confident observations. Orthologs of RING1 and PCGF proteins that make the heterodimeric PRC1 core module, on the one hand, and orthologs of EZH, EED, SUZ12 and RBBP of the PRC2 core module, on the other, were found in, roughly, 40% and 50% of the species belonging to all major groups.

The results are consistent with the presence of both PRC1 and PRC2 in the last eukaryotic common ancestor (LECA), thus extending the evolutionary story of the Polycomb system. A clear tendency to the maintenance/loss of core subunits for the same complex is shown, while distributions of PRC1 and PRC2 orthologs indicate separate evolution. In this regard, the work provides evolutionary support for recent evidence showing uncoupling of extant PRC1 and PRC2 activities. It also confirms that the current enzymatic core of PRC1 appeared unrelated to a writer activity on histone H2A. It may have been interesting to investigate in the orthologs of the RING1 moiety the possible conservation of sequences known to interact with the E2 ligase to speculate about the late ubiquitylation of H2A.

Subsequently, the authors extend their work into PRC1 subunits, to conclude that paralogs of RYBP and CBX, subunits that bind, in a mutually exclusive manner, to the RING1 component of the PRC1 core complex, share a common ancestor. RYBP-containing PRC1 complexes are known as non-canonical or variant (ncPRC1/vPRC1), whereas those containing CBX paralogs are canonical PRC1 (cPRC1, identified first and fitting initially known Polycomb functions). Typical cPRC1 subunits include the PHC paralogs. CBX paralogs have a chromatin reader module (chromodomain, recognizing H3K27me3) and PHC paralogs a protein-protein interacting module (SAM domain), thought to be important for the formation of high-order chromatin structures. The distribution of potential orthologs for these domains of cPRC1 subunits, more recent than that of RYBP paralogs, leads the authors to conclude an earlier appearance of ncPRC1 complexes. However, this suggestion would be more robust if a concurrent distribution of RYBP paralogs together with that of core PRC1 subunits was shown, so that as the togetherness of RING and PCGF paralogs might be extended to them too. I also wonder, whether a description of ancient proteins sharing the so-called CBX_7 domain might support better the suggested common origin of RYBP and CBX products.

Some aspects for improvement of the Introduction (no impact on reported findings):

- Quotation of previous work: too often reviews are used (lines 72, 82, 84, 87) instead of original papers. Occasionally, cited literature (line 82) does not accurately support what it is meant to support. There are clear omissions too.
- Recent findings showing an unexpected relevance for PRC1-dependent modification of H2A in mammalian embryonic stem cells (Klose's lab), in opposition to a preferred chromatin compaction model for Polycomb repression, are not appropriately reflected. It is likely that both are used in a context-dependent manner.
- Non-canonical PRC1 complexes are also known as variant PRC1 complexes. This should be indicated, for clarity, even if the ncPRC1 option is subsequently used exclusively.
- In a similar line, include a definition/reference for zf-RING (line 76), usually known as RING finger domain.
- An abbreviation list for domains, proteins, may be helpful.

Reviewer #3 (Remarks to the Author):

This study explores the deep conservation of PRC1 in eukaryotes. They use manually curated sequence alignment and Alphfold 2 to identify orthologs of the core subunits of PRC2 even in species where they were not previously found. They also identify orthologs of PRC1 core subunit RING1 and PCGF in > 67 species (out of 178 species sampled to represent evenly the various groups of eukaryotes. Importantly orthologs are identified in protists that are placed proximal to the root of the eukaryotic tree, leading to conclude to the conservation of PRC1 and PRC2 in LECA. PRC1 subunit were also found in fungi and diatoms, where they were considered absent. While the gain/loss of core subunits of PRC1 and PRC2 were coupled, the study reveals the unexpected absence of coupling of evolution of PRC1 and PRC2.

Although the subunits CBX8 and RYBP defines the canonical and non canonical PRC1 complexes, the Authors propose that CBX8 and RYBP are distantly related paralogs that share a beta hairpin, based on structural similarity. CBX is identified only in animals while a distant ortholog of RYBP is reported not only in choanoflagellates but also in a distant relative of fungi. The authors propose that RYPB preexisted to CBX. Importantly the study outlines that the H2AKUb by PRC1 was acquired quite recently implying that the cross talk between PRC1 and PRC2 evolved avec their selection and in many groups PRC1 or PRC2 alone independently evolved to more specialized complexes and roles. Most of the work rests on solid analysis and sheds light on interesting and unexpected findings. The figures clearly illustrate the findings.

Major points:

- 1.The conservation of PRC1 and PRC2 core subunits in a larger fraction of the eukaryotic species studied is remarkable but the Authors make no comment as why the subunits of PRC1 or PRC2 were not found in a large number of species surveyed (>100 in the case of PRC1).
- 2.On the same line if the absence of detection relates to incomplete or improper genome assembly, one might rather expect – contrary to the author’s expectation that further analyses will reveal more coupling between PRC1 and PRC2. Could the Authors discuss this point in more details.
3. The conclusion of the earlier origin of RYPB than CBX rests on very little species and it would be important to increase the support or tone the conclusion down.

Minor points

- 1.The introduction is very metazoan centric and could add a few statements regarding the role of Polycomb repressive complexes in plants, particularly its role in development PMID: 26313233, PMID: 25449722 Review and responses to environmental changes PMID: 29920687, and also cite works in fungi PMID: 28196760.
- 2.The statements lines 143- 150 are redundant.
- 3.Line 238 the statement needs correction “Indeed, we initially missed a *Cryptococcus neoformans* SUZ12 ortholog the yeast *Cryptococcus neoformans*.”

Reviewer #1 (Remarks to the Author):

Here the authors present a comprehensive analysis of the evolution of the Polycomb group proteins from the Last Eukaryotic Common Ancestor throughout the eukaryotic tree, by looking for the presence of the main PRC1 and PRC2 core components in diverse eukaryotes. Their approach (including manual curation of the domains eg RING, RAUWL, VEFS , to avoid the lack of sensitivity from default Pfam profiles) has led to the identification of high-confidence orthologs including ones missed in previous studies and in organisms thought to have lost orthologs. The authors show that over evolution the core components within a PRC complex are either lost or retained together, but between the complexes are uncoupled, with many organisms only having either PRC1 or PRC2. The authors also show that RYBP (in ncPRC1) appears to have originated before the structurally similar CBX protein (in cPRC1). Taken together, this suggests an independent function of PRC1 and PRC2 and that the crosstalk between complexes, especially CBX binding to K27me3, is a secondary development in evolution.

The work presented here is a very thorough search for presence of PRC1 and PRC2 core components throughout evolution of eukaryotes. Using manual curation by searching alignments of the known protein domains, rather than just relying on Pfam profiles, has shown the presence of PRC1 and PRC2 (and indeed loss in some lineages) in more organisms than previously thought which is very valuable and clears up the literature. It is especially interesting when thinking about the mechanism of repression of these complexes and suggests that the two protein complexes can work independently. In terms of the models of recruitment to chromatin, the finding that the ncPRC1 complex evolved before cPRC1, adds weight to the argument that PRC1 can work independently of PRC2 and can be recruited without the need of CBX proteins to bind to the H3K27me3 of PRC2. This is an important observation.

Concern 1: In humans, there are 6 PCGF proteins, and indeed the authors look at the phylogenetic tree of PCGF protein in FigS1b, although when each of these different proteins evolved wasn't discussed in the text. Which type of human PCGF protein did the ancestral one look most like? This would be an interesting point to expand on, particularly because the type of PCGF protein can also determine if the PRC1 complex is canonical or non-canonical.

Response 1: We appreciate the reviewer's interest in this aspect of our research and agree that our results could be used to address this interesting question of which human PCGF is most similar to the ancestral PCGF protein.

Action 1: To address this point, we re-evaluated our phylogenetic analyses and domain comparisons to provide a more detailed account of the ancestral PCGF protein's resemblance to specific human PCGF variants. Similar to a recent paper in PNAS by Gahan et al. on the evolution of PRC1 (Gahan et al., 2020), one could say that that the canonical *PCGF2/4* and the non-canonical *PCGF1, 3, 5, and 6* genes form monophyletic sister groups as a result of a duplication at the root of metazoa. Our findings however also suggest the possibility that subsequent duplications within *PCGF1, 3, 5, and 6* also took place at the root of metazoans, given that multiple homologs (such as those found in *Nematostella vectensis* (NEMVEC), *Branchiostoma belcheri* (BRABEL), and *Crassostrea gigas* (CRAGIG) within that cluster, cluster distantly. We also found PCGF in the sister group of animals (choanoflagellates) to be recurrently annotated as PCGF5 by eggNOG, implying a possibly greater sequence similarity of these sequences to the PCGF5 clade over other PCGF paralogs.

Even though all these observations are very interesting, we concluded there is no clear evidence to show whether a definite non-canonical or canonical PCGF variant was present at the root of metazoans. Instead, we propose that the actual diversification and functional differences between non-canonical PCGF and canonical PCGF likely emerged after the duplications we mentioned earlier. Consequently, we believe the ancestral PCGF should not be considered either more canonical or non-canonical.

In light of these findings, we have updated the results of our manuscript and incorporated a new paragraph to discuss these aspects of our data. See lines 184-199 in the revised manuscript.

Concern 2: The authors also only looked at the core components of PRC2 but it would be interesting to either look for the presence of the accessory proteins eg Jarid2, Aebp2 or PCL proteins, or in the least add a comment to the text about the evolution of the different types of PRC2 complexes (in a similar way to ncPRC1 and cPRC1).

Response 2: The reviewer is correct that for general interest and to enrich our understanding of PRC2 evolution, it could be relevant to include a section detailing the evolution of different subtypes of PRC2 complexes.

Action 2: In the context of the present manuscript, we conducted a preliminary investigation into these proteins, but it revealed very complex homology, duplication, family and orthology relations, due to their additional functions beyond their role in PRC2 recruitment. This makes it a novel research line for a detailed study in a future report. Yet, we revised the introduction to include a comment on the evolution of different subtypes of PRC2 complexes.

Line78: Beyond its canonical core subunits, PRC2 engages in interactions with an array of accessory proteins. These accessory subunits, including but not limited to JARID2, AEBP2, and PCL, associate with the PRC2 core, leading to the formation of distinct subassemblies. These subassemblies perform specialized functions, allowing PRC2 to adapt to varying cellular contexts

Reviewer #2 (Remarks to the Author):

The manuscript by de Potter et al. is a study on the phylogenetic of Polycomb proteins. Exploiting the growing collection of eukaryotic genomes the authors search throughout 178 species for possible evolutionary ancestors of core subunits of the two major classes of Polycomb Repressive Complexes, PRC1 and PRC2. It extends significantly previous observations on Polycomb phylogeny resulting from more detailed work adding new information of interest to researchers in the field and, in particular, to those attracted by molecular evolution.

PRC core complexes make enzymatic modules responsible for histone modifications: a heterodimeric ubiquitin E3 ligase for lysine 119 of histone H2A in PRC1, and the SET domain lysine methyltransferase for lysine 27 of histone H3 (H3K27me3), plus accessory subunits, in PRC2. Searching for combinations of protein motifs found in core subunits results in meaningful, confident observations. Orthologs of RING1 and PCGF proteins that make the heterodimeric PRC1 core module, on the one hand, and orthologs of EZH, EED, SUZ12 and RBBP of the PRC2 core module, on the other, were found in, roughly, 40% and 50% of the species belonging to all major groups.

The results are consistent with the presence of both PRC1 and PRC2 in the last eukaryotic common ancestor (LECA), thus extending the evolutionary story of the Polycomb system. A clear tendency to the maintenance/loss of core subunits for the same complex is shown, while distributions of PRC1 and PRC2 orthologs indicate separate evolution. In this regard, the work provides evolutionary support for recent evidence showing uncoupling of extant PRC1 and PRC2 activities. It also confirms that the current enzymatic core of PRC1 appeared unrelated to a writer activity on histone H2A.

Concern 3: It may have been interesting to investigate in the orthologs of the RING1 moiety the possible conservation of sequences known to interact with the E2 ligase to speculate about the late ubiquitylation of H2A.

Response 3: We appreciate the reviewer's suggestion to investigate the possible conservation of sequences known to interact with the E2 ligase to speculate about the late ubiquitylation of H2A. To our knowledge, the exact residues in RING1 responsible for the interaction with the E2 ligase are not yet known.

Action 3: We have therefore looked at the ring domains of CNOT4 and BRCA1 and the mode of interaction with their respective E2 ligase partners, as for these, there is information on the residues crucial for establishing a functional interaction (see (Albert et al., 2002; Brzovic et al., 2003), respectively). These ring-E2 interactions both involve bulky hydrophobic residues located between the first pair of cysteine residues of the ring domain. We checked the alignment of our curated set of RING1 orthologs produced in this study, and identified indeed a conserved, somewhat bulky, hydrophobic residue on this exact position in the form of isoleucine or valine at alignment position 757 (position 50 of human RING) (see a screenshot of the raw alignment attached below). This opens up the possibility that the large majority of RING1 orthologs identified here maintain their ability to interact functionally with an E2 ligase (most likely UbE2E1, see: (Wheaton et al., 2017) and therefore possibly are capable of catalysing H2AK119 ubiquitination. However, this of course is purely speculative and hinges on the assumption that this residue is always biochemically active. It is thus difficult to draw

conclusions from this data, especially in the light of the absence of experimental evidence for H2AK119ub in non-holozoan or plant species shown by (Grau-Bové et al., 2022).

Subsequently, the authors extend their work into PRC1 subunits, to conclude that paralog of RYBP and CBX, subunits that bind, in a mutually exclusive manner, to the RING1 component of the PRC1 core complex, share a common ancestor. RYBP-containing PRC1 complexes are known as non-canonical or variant (ncPRC1/vPRC1), whereas those containing CBX paralog are canonical PRC1 (cPRC1, identified first and fitting initially known Polycomb functions). Typical cPRC1 subunits include the PHC paralog. CBX paralog has a chromatin reader module (chromodomain, recognizing H3K27me3) and PHC paralog a protein-protein interacting module (SAM domain), thought to be important for the formation of high-order chromatin structures.

Concern 4: The distribution of potential orthologs for these domains of cPRC1 subunits, more recent than that of RYBP paralog, leads the authors to conclude an earlier appearance of ncPRC1 complexes. However, this suggestion would be more robust if a concurrent distribution of RYBP paralog together with that of core PRC1 subunits was shown, so that as the togetherness of RING and PCGF paralog might be extended to them too. I also wonder, whether a description of ancient proteins sharing the so-called CBX_7 domain might support better the suggested common origin of RYBP and CBX products.

Response 4: We recognize the potential of such an addition to further strengthen our hypothesis regarding the shared origin of RYBP and CBX proteins. While the notion of establishing a co-distribution of RYBP paralog alongside core PRC1 subunits or identifying other ancient proteins with the CBX_7 domain was in line with our research aims, we unfortunately encountered limitations in achieving additional evidence for these points. We made efforts to conduct several phylogenetic analyses pre-submission, involving the

chromobox domain, chromoshadow domain, and CBX7_C domain, and we explored the N-terminal Znf-RanBP domain of RYBP. Unfortunately, we faced challenges in working with such multifunctional (present in many different proteins) and compact domains (~16 AA).

Action 4: We have reevaluated our results and rephrased our conclusions to reflect a more cautious tone.

In line 138 changed: Furthermore, we discovered previously unreported ncPRC1-defining subunits in the relatives of animals and fungi, **suggesting** that the differentiation of ncPRC1 occurred before cPRC1.

In line 345 changed: 'Based on this evidence, we **propose** that the domains CBX7_C and YAF are **might** be distant paralogs.'

In line 349 changed: 'Combined, the structural similarity and the reciprocal best hits of their β -hairpin domains are first indications that CBX and RYBP might be switching paralogs in PRC1.'

In line 421 changed: 'Moreover, we **propose** that the origin of ncPRC1 predates that of cPRC1, adding weight earlier hypotheses that ncPRC1 accessory subunits originated before cPRC1².'

Some aspects for improvement of the Introduction (no impact on reported findings):

Concern 5: Quotation of previous work: too often reviews are used (lines 72, 82, 84, 87) instead of original papers. Occasionally, cited literature (line 82) does not accurately support what it is meant to support. There are clear omissions too.

Response and action 5: We thank the reviewer for carefully revising the manuscript in this regard. We have now adjusted the literature.

In line 84 (previously 72) added: (Hauri et al., 2016; Lagarou et al., 2008)

In line 94 (previously 82) added: (de Napoles et al., 2004; Gao et al., 2012; Tavares et al., 2012)

In line 96 (previously 84) added: (Blackledge et al., 2014, 2020; Cohen et al., 2021; Fursova et al., 2019; Leeb et al., 2010; Pengelly et al., 2015; Scelfo et al., 2019; Sijm et al., 2022; Tamburri et al., 2020; Tavares et al., 2012; Wang et al., 2004; Zepeda-Martinez et al., 2020)

In line 97 (previously 87) added: (Blackledge et al., 2014, 2020; Bonnet et al., 2022; Cohen et al., 2021; Fursova et al., 2019; Leeb et al., 2010; Pengelly et al., 2015; Scelfo et al., 2019; Sijm et al., 2022; Tamburri et al., 2020; Tavares et al., 2012; Wang et al., 2004; Zepeda-Martinez et al., 2020)

In line 98 added: Studies in mouse embryonic stem cells have indicated that H2Aub1 is central to gene repression by the Polycomb system (Blackledge et al., 2014, 2020; Tavares et al., 2012). However, other studies in mouse embryonic stem cells revealed redundancy between H2Aub1 and H3K27me3 (Cohen et al., 2021; Leeb et al., 2010; Zepeda-Martinez et al., 2020). Although H2Aub1 is essential for animal viability, mutational studies established that PRC1 rather than its enzymatic activity is essential for transcriptional repression of canonical Polycomb target genes (Bonnet et al., 2022; Illingworth et al., 2015; Pengelly et al., 2015; Tsuboi et al., 2018).

Concern 6: Recent findings showing an unexpected relevance for PRC1-dependent modification of H2A in mammalian embryonic stem cells (Klose's lab), in opposition to a preferred chromatin compaction model for Polycomb repression, are not appropriately reflected. It is likely that both are used in a context-dependent manner.

Response and action 6: We have now included these findings into our manuscript (see references in **Concern 5**)

Concern 7:- Non-canonical PRC1 complexes are also known as variant PRC1 complexes. This should be indicated, for clarity, even if the ncPRC1 option is subsequently used exclusively.

Response and action 7: We thank the reviewer for this comment, and for carefully revising the manuscript in this regard. We have made the necessary adjustment to reflect this terminology.

Line 91: The presence of a RYBP/YAF subunit defines the ncPRC1s (also known as variant PRC1 complexes),

Concern 8:-In a similar line, include a definition/reference for zf-RING (line 76), usually known as RING finger domain.

Response and action 8: Thank you for this suggestion, we have included a definition at the specified location in our manuscript.

Line 87: Notably, plant RING1 and PCGF orthologs can form homo- as well as heterodimers⁴. RING1 and PCGF proteins share a similar structural organization characterized by an N-terminal zf-RING domain (usually known as RING finger domain) and a C-terminal RAWUL domain (**Fig.1B**)

Concern 9: An abbreviation list for domains, proteins, may be helpful.

Response and action 9: We have made efforts to more specifically indicate this in the description of Figure1:

Abbreviations: RAWUL: RING finger- and WD40-associated ubiquitin-like, MCSS: motif connecting SANT1 and SANT2, SANT: SWI3, ADA2, N-CoR and TFIIB DNA-binding domain, CxC: cysteine-rich domain, SET: Su(var) 3–9, enhancer of zeste, trithorax domain, WD: WD-40 domain, WDB: WD-40 binding domain, Zn: Zn-finger region, VEFS:VRN2-EMF2-FIS2-SU(Z)12.

Reviewer #3 (Remarks to the Author):

This study explores the deep conservation of PRC1 in eukaryotes. They use manually curated sequence alignment and Alphfold 2 to identify orthologs of the core subunits of PRC2 even in species where they were not previously found. They also identify orthologs of PRC1 core subunit RING1 and PCGF in > 67 species (out of 178 species sampled to represent evenly the various groups of eukaryotes). Importantly orthologs are identified in protists that are placed proximal to the root of the eukaryotic tree, leading to conclude to the conservation of PRC1 and PRC2 in LECA. PRC1 subunit were also found in fungi and diatoms, where they were considered absent. While the gain/loss of core subunits of PRC1 and PRC2 were coupled, the study reveals the unexpected absence of coupling of evolution of PRC1 and PRC2.

Although the subunits CBX8 and RYBP defines the canonical and non canonical PRC1 complexes, the Authors propose that CBX8 and RYBP are distantly related paralogs that share a beta hairpin, based on structural similarity. CBX is identified only in animals while a distant ortholog of RYBP is reported not only in choanoflagellates but also in a distant relative of fungi. The authors propose that RYPB preexisted to CBX. Importantly the study outlines that the H2AKUb by PRC1 was acquired quite recently implying that the cross talk between PRC1 and PRC2 evolved avec their selection and in many groups PRC1 or PRC2 alone independently evolved to more specialized complexes and roles.

Most of the work rests on solid analysis and sheds light on interesting and unexpected findings. The figures clearly illustrate the findings.

Major points:

Concern 10:-The conservation of PRC1 and PRC2 core subunits in a larger fraction of the eukaryotic species studied is remarkable but the Authors make no comment as why the subunits of PRC1 or PRC2 were not found in a large number of species surveyed (>100 in the case of PRC1).

Response 10: We thank the reviewer for their comment, which brings attention to an important aspect of our study. We were perhaps not as surprised as we know from extensive evolutionary genomic studies that many systems which are thought of as essential in mammalian systems turn out to have patchy presence profiles (ska, minor spliceosome, cilia) and recent studies stress the importance of gene loss in genome evolution (Albalat & Cañestro, 2016; Eliáš et al., 2016; Russell et al., 2006). Additionally, the high number of genomes without PRC1 and/or PRC2 compared to other recent studies also reflects our inclusion of recently sequenced eukaryotic microbial biodiversity. We agree though that the high number of genomes without PRC1 or PRC2 merits some more emphasis, contextualization and reflection both from a biological as well as technical perspective.

Action 10: We have now taken steps to address this in our manuscript and have added text to the '*Intra-complex evolution of PRC1 and PRC2 is cohesive while inter-complex evolution is uncoupled*' paragraph to include this.

Line 308: The evolutionary flexibility of the system is similarly reflected in the large number of species that lack PRC1 or PRC2. While the possibility exists that instances where the detection of these complexes was missed could be associated with incomplete or inaccurate genome assembly, the loss of a crucial system such as Polycomb aligns with observations of loss in other crucial molecular mechanisms across different organisms. Notable instances include the ska system, minor spliceosome, and cilia, which all have an uneven distribution among species due to the influence of gene loss pressures throughout genome evolution (Albalat & Cañestro, 2016; Eliáš et al., 2016; Russell et al., 2006; van Hooff et al., 2017). In

fact, recent research has also brought to light that in Plasmodium, the experimentally verified absence of PRCs has led cohesin to perform the role of gene regulation, highlighting the potential of alternative mechanisms substituting the Polycomb system (Rosa et al., 2023).

Concern 11: On the same line if the absence of detection relates to incomplete or improper genome assembly, one might rather expect – contrary to the author’s expectation that further analyses will reveal more coupling between PRC1 and PRC2. Could the Authors discuss this point in more detail?

Response 11: We thank the reviewer for this observation, with which we are very familiar and for which we had integrated quite some precaution to minimize the impact of incomplete or otherwise noisy genome assembly quality. We regret that we did not include our efforts with sufficient clarity in the manuscript. We have taken the opportunity to address this point with greater emphasis and more specific detail in our revised manuscript. We recognize that the absence of detection could indeed raise questions about the potential for further coupling between PRC1 and PRC2. While we have made efforts to reduce the possibility of missing orthologs due to sequence divergence or assembly issues, it is important to acknowledge that no method is entirely immune to such limitations.

Action 11: We have reassessed our methods and the robustness of our eukaryotic database compilation and are confident in the soundness of our approach. In line with concern 10, we have adjusted our manuscript and included a paragraph in which we recognize this limitation. In addition, we have refined the “Materials and Methods” section to improve the clarity of our efforts and added a line to the discussion to take this concern into consideration.

Line 418: . Incomplete or inaccurate genome assemblies may have also falsely suggested the absence of certain Polycomb complexes, and led to an over-estimation of uncoupled evolution. However, we have made extensive efforts to enhance the quality of our eukaryotic database, thereby minimizing the chance that we falsely infer absences, (SI Data and Methods).

Concern 12: The conclusion of the earlier origin of RYPB than CBX rests on very little species and it would be important to increase the support or tone the conclusion down.

Response 12: We appreciate the reviewer’s point highlighting the basis of our conclusion and the importance of either increasing the support or toning the conclusion down.

Action 12: In line with the input from Reviewer 2 (**See Concern 4**), we have taken steps to adjust the corresponding sections in our manuscript and for clarification better differentiate our observations (presence of ~RYBP like in early branching) and the potential conclusion for origin (which we now toned down).

In line 138 changed: Furthermore, we discovered previously unreported ncPRC1-defining subunits in the relatives of animals and fungi, **suggesting** that the differentiation of ncPRC1 occurred before cPRC1.

In line 345 changed: ‘Based on this evidence, we **propose** that the domains CBX7_C and YAF are **might** be distant paralogs.’

In line 349 changed: ‘Combined, the structural similarity and the reciprocal best hits of their β -hairpin domains are first indications that CBX and RYBP might be switching paralogs in PRC1.’

In line 425 changed: ‘Moreover, we **propose** that the origin of ncPRC1 predates that of cPRC1, adding weight earlier hypotheses that ncPRC1 accessory subunits originated before cPRC1 ².’

Minor points

1.The introduction is very metazoan centric and could add a few statements regarding the role of Polycomb repressive complexes in plants, particularly its role in development PMID: 26313233, PMID: 25449722 Review and responses to environmental changes PMID: 29920687, and also cite works in fungi PMID: 28196760.

We have added the following lines with references to line 108 of the manuscript:
Thus, while the molecular analysis of mechanisms of Polycomb repression have been based mainly on animal cells, it is important to realize that the Polycomb system might function differently in plants and non-metazoan organisms. The Polycomb system plays a crucial role in the developmental regulation of plants and in their response to environmental changes (Friedrich et al., 2019; Gan et al., 2015; Xiao & Wagner, 2015). Moreover, there is accumulating evidence that the Polycomb system has diverse roles in genome regulation in fungi (Lewis, 2017; Ridenour et al., 2020)

2.The statements lines 143- 150 are redundant.

We have now made efforts to reduce the redundancy in line 150-159.

3.Line 238 the statement needs correction “Indeed, we initially missed a *Cryptococcus neoformans* SUZ12 ortholog the yeast *Cryptococcus neoformans*.”

We thank the reviewer for noticing this mistake, we have adjusted the manuscript (line 256 now)

References

- Albalat, R., & Cañestro, C. (2016). Evolution by gene loss. *Nature Reviews. Genetics*, 17(7), 379–391. <https://doi.org/10.1038/nrg.2016.39>
- Albert, T. K., Hanzawa, H., Legtenberg, Y. I. A., de Ruwe, M. J., van den Heuvel, F. A. J., Collart, M. A., Boelens, R., & Timmers, H. T. M. (2002). Identification of a ubiquitin-protein ligase subunit within the CCR4-NOT transcription repressor complex. *The EMBO Journal*, 21(3), 355–364. <https://doi.org/10.1093/emboj/21.3.355>
- Blackledge, N. P., Farcas, A. M., Kondo, T., King, H. W., McGouran, J. F., Hanssen, L. L. P., Ito, S., Cooper, S., Kondo, K., Koseki, Y., Ishikura, T., Long, H. K., Sheahan, T. W., Brockdorff, N., Kessler, B. M., Koseki, H., & Klose, R. J. (2014). Variant PRC1 Complex-Dependent H2A Ubiquitylation Drives PRC2 Recruitment and Polycomb Domain Formation. *Cell*, 157(6), 1445–1459. <https://doi.org/10.1016/j.cell.2014.05.004>
- Blackledge, N. P., Fursova, N. A., Kelley, J. R., Huseyin, M. K., Feldmann, A., & Klose, R. J. (2020). PRC1 Catalytic Activity Is Central to Polycomb System Function. *Molecular Cell*, 77(4), 857-874.e9. <https://doi.org/10.1016/j.molcel.2019.12.001>
- Bonnet, J., Boichenko, I., Kalb, R., Le Jeune, M., Maltseva, S., Pieropan, M., Finkl, K., Fierz, B., & Müller, J. (2022). PR-DUB preserves Polycomb repression by preventing excessive accumulation of H2Aub1, an antagonist of chromatin compaction. *Genes & Development*, 36(19–20), 1046–1061. <https://doi.org/10.1101/gad.350014.122>

- Brzovic, P. S., Keefe, J. R., Nishikawa, H., Miyamoto, K., Fox, D., Fukuda, M., Ohta, T., & Kleit, R. (2003). Binding and recognition in the assembly of an active BRCA1/BARD1 ubiquitin-ligase complex. *Proceedings of the National Academy of Sciences of the United States of America*, *100*(10), 5646–5651.
<https://doi.org/10.1073/pnas.0836054100>
- Cohen, I., Bar, C., Liu, H., Valdes, V. J., Zhao, D., Galbo, P. M., Silva, J. M., Koseki, H., Zheng, D., & Ezhkova, E. (2021). Polycomb complexes redundantly maintain epidermal stem cell identity during development. *Genes & Development*, *35*(5–6), 354–366.
<https://doi.org/10.1101/gad.345363.120>
- de Napoles, M., Mermoud, J. E., Wakao, R., Tang, Y. A., Endoh, M., Appanah, R., Nesterova, T. B., Silva, J., Otte, A. P., Vidal, M., Koseki, H., & Brockdorff, N. (2004). Polycomb Group Proteins Ring1A/B Link Ubiquitylation of Histone H2A to Heritable Gene Silencing and X Inactivation. *Developmental Cell*, *7*(5), 663–676.
<https://doi.org/10.1016/j.devcel.2004.10.005>
- Eliáš, M., Klimeš, V., Derelle, R., Petřelková, R., & Tachezy, J. (2016). A pan-eukaryotic genomic analysis of the small GTPase RABL2 underscores the significance of recurrent gene loss in eukaryote evolution. *Biology Direct*, *11*(1), 5.
<https://doi.org/10.1186/s13062-016-0107-8>
- Friedrich, T., Faivre, L., Bäurle, I., & Schubert, D. (2019). Chromatin-based mechanisms of temperature memory in plants. *Plant, Cell & Environment*, *42*(3), 762–770.
<https://doi.org/10.1111/pce.13373>
- Fursova, N. A., Blackledge, N. P., Nakayama, M., Ito, S., Koseki, Y., Farcas, A. M., King, H. W., Koseki, H., & Klose, R. J. (2019). Synergy between Variant PRC1 Complexes Defines Polycomb-Mediated Gene Repression. *Molecular Cell*, *74*(5), 1020–1036.e8.
<https://doi.org/10.1016/j.molcel.2019.03.024>
- Gahan, J. M., Rentzsch, F., & Schnitzler, C. E. (2020). The genetic basis for PRC1 complex diversity emerged early in animal evolution. *Proceedings of the National Academy of Sciences*, *117*(37), 22880–22889. <https://doi.org/10.1073/pnas.2005136117>
- Gan, E.-S., Xu, Y., & Ito, T. (2015). Dynamics of H3K27me3 methylation and demethylation in plant development. *Plant Signaling & Behavior*, *10*(9), e1027851.
<https://doi.org/10.1080/15592324.2015.1027851>
- Gao, Z., Zhang, J., Bonasio, R., Strino, F., Sawai, A., Parisi, F., Kluger, Y., & Reinberg, D. (2012). PCGF homologs, CBX proteins, and RYBP define functionally distinct PRC1 family complexes. *Molecular Cell*, *45*(3), 344–356.
<https://doi.org/10.1016/j.molcel.2012.01.002>
- Grau-Bové, X., Navarrete, C., Chiva, C., Pribasnik, T., Antó, M., Torruella, G., Galindo, L. J., Lang, B. F., Moreira, D., López-García, P., Ruiz-Trillo, I., Schleper, C., Sabidó, E., & Sebé-Pedrós, A. (2022). A phylogenetic and proteomic reconstruction of eukaryotic chromatin evolution. *Nature Ecology & Evolution*, *6*(7), 1007–1023.
<https://doi.org/10.1038/s41559-022-01771-6>
- Hauri, S., Comoglio, F., Seimiya, M., Gerstung, M., Glatzer, T., Hansen, K., Aebbersold, R., Paro, R., Gstaiger, M., & Beisel, C. (2016). A High-Density Map for Navigating the Human Polycomb Complexome. *Cell Reports*, *17*(2), 583–595.
<https://doi.org/10.1016/j.celrep.2016.08.096>
- Illingworth, R. S., Moffat, M., Mann, A. R., Read, D., Hunter, C. J., Pradeepa, M. M., Adams, I. R., & Bickmore, W. A. (2015). The E3 ubiquitin ligase activity of RING1B is not essential for early mouse development. *Genes & Development*, *29*(18), 1897–1902.
<https://doi.org/10.1101/gad.268151.115>

- Lagarou, A., Mohd-Sarip, A., Moshkin, Y. M., Chalkley, G. E., Bezstarosti, K., Demmers, J. A. A., & Verrijzer, C. P. (2008). dKDM2 couples histone H2A ubiquitylation to histone H3 demethylation during Polycomb group silencing. *Genes & Development*, *22*(20), 2799–2810. <https://doi.org/10.1101/gad.484208>
- Leeb, M., Pasini, D., Novatchkova, M., Jaritz, M., Helin, K., & Wutz, A. (2010). Polycomb complexes act redundantly to repress genomic repeats and genes. *Genes & Development*, *24*(3), 265–276. <https://doi.org/10.1101/gad.544410>
- Lewis, Z. A. (2017). Polycomb Group Systems in Fungi: New Models for Understanding Polycomb Repressive Complex 2. *Trends in Genetics : TIG*, *33*(3), 220–231. <https://doi.org/10.1016/j.tig.2017.01.006>
- Pengelly, A. R., Kalb, R., Finkl, K., & Müller, J. (2015). Transcriptional repression by PRC1 in the absence of H2A monoubiquitylation. *Genes & Development*, *29*(14), 1487–1492. <https://doi.org/10.1101/gad.265439.115>
- Ridenour, J. B., Möller, M., & Freitag, M. (2020). Polycomb Repression without Bristles: Facultative Heterochromatin and Genome Stability in Fungi. *Genes*, *11*(6). <https://doi.org/10.3390/genes11060638>
- Rosa, C., Singh, P., Chen, P., Sinha, A., Claës, A., Preiser, P. R., Dedon, P. C., Baumgarten, S., Scherf, A., & Bryant, J. M. (2023). Cohesin contributes to transcriptional repression of stage-specific genes in the human malaria parasite. *EMBO Reports*, e57090. <https://doi.org/10.15252/embr.202357090>
- Russell, A. G., Charette, J. M., Spencer, D. F., & Gray, M. W. (2006). An early evolutionary origin for the minor spliceosome. *Nature*, *443*(7113), 863–866. <https://doi.org/10.1038/nature05228>
- Scelfo, A., Fernández-Pérez, D., Tamburri, S., Zanotti, M., Lavarone, E., Soldi, M., Bonaldi, T., Ferrari, K. J., & Pasini, D. (2019). Functional Landscape of PCGF Proteins Reveals Both RING1A/B-Dependent-and RING1A/B-Independent-Specific Activities. *Molecular Cell*, *74*(5), 1037-1052.e7. <https://doi.org/10.1016/j.molcel.2019.04.002>
- Sijm, A., Atlasi, Y., van der Knaap, J. A., Wolf van der Meer, J., Chalkley, G. E., Bezstarosti, K., Dekkers, D. H. W., Doff, W. A. S., Ozgur, Z., van IJcken, W. F. J., Demmers, J. A. A., & Verrijzer, C. P. (2022). USP7 regulates the ncPRC1 Polycomb axis to stimulate genomic H2AK119ub1 deposition uncoupled from H3K27me3. *Science Advances*, *8*(44). <https://doi.org/10.1126/sciadv.abq7598>
- Tamburri, S., Lavarone, E., Fernández-Pérez, D., Conway, E., Zanotti, M., Manganaro, D., & Pasini, D. (2020). Histone H2AK119 Mono-Ubiquitination Is Essential for Polycomb-Mediated Transcriptional Repression. *Molecular Cell*, *77*(4), 840-856.e5. <https://doi.org/10.1016/j.molcel.2019.11.021>
- Tavares, L., Dimitrova, E., Oxley, D., Webster, J., Poot, R., Demmers, J., Bezstarosti, K., Taylor, S., Ura, H., Koide, H., Wutz, A., Vidal, M., Elderkin, S., & Brockdorff, N. (2012). RYBP-PRC1 Complexes Mediate H2A Ubiquitylation at Polycomb Target Sites Independently of PRC2 and H3K27me3. *Cell*, *148*(4), 664–678. <https://doi.org/10.1016/j.cell.2011.12.029>
- Tsuboi, M., Kishi, Y., Yokozeki, W., Koseki, H., Hirabayashi, Y., & Gotoh, Y. (2018). Ubiquitination-Independent Repression of PRC1 Targets during Neuronal Fate Restriction in the Developing Mouse Neocortex. *Developmental Cell*, *47*(6), 758-772.e5. <https://doi.org/10.1016/j.devcel.2018.11.018>
- van Hooff, J. J., Tromer, E., van Wijk, L. M., Snel, B., & Kops, G. J. (2017). Evolutionary dynamics of the kinetochore network in eukaryotes as revealed by comparative

- genomics. *EMBO Reports*, 18(9), 1559–1571.
<https://doi.org/10.15252/embr.201744102>
- Wang, H., Wang, L., Erdjument-Bromage, H., Vidal, M., Tempst, P., Jones, R. S., & Zhang, Y. (2004). Role of histone H2A ubiquitination in Polycomb silencing. *Nature*, 431(7010), 873–878. <https://doi.org/10.1038/nature02985>
- Wheaton, K., Sarkari, F., Stanly Johns, B., Davarinejad, H., Egorova, O., Kaustov, L., Raught, B., Saridakis, V., & Sheng, Y. (2017). UbE2E1/UBCH6 Is a Critical in Vivo E2 for the PRC1-catalyzed Ubiquitination of H2A at Lys-119. *The Journal of Biological Chemistry*, 292(7), 2893–2902. <https://doi.org/10.1074/jbc.M116.749564>
- Xiao, J., & Wagner, D. (2015). Polycomb repression in the regulation of growth and development in Arabidopsis. *Current Opinion in Plant Biology*, 23, 15–24.
<https://doi.org/10.1016/j.pbi.2014.10.003>
- Zepeda-Martinez, J. A., Pribitzer, C., Wang, J., Bsteh, D., Golumbeanu, S., Zhao, Q., Burkard, T. R., Reichholf, B., Rhie, S. K., Jude, J., Moussa, H. F., Zuber, J., & Bell, O. (2020). Parallel PRC2/cPRC1 and vPRC1 pathways silence lineage-specific genes and maintain self-renewal in mouse embryonic stem cells. *Science Advances*, 6(14).
<https://doi.org/10.1126/sciadv.aax5692>

REVIEWERS' COMMENTS:

Reviewer #1 (Remarks to the Author):

The authors have added in an additional analysis of the PCGF proteins which answers the questions I had and I am happy that it addresses my point. They have also added into the introduction some information about the different PRC2 complexes and done some preliminary analysis to suggest that this area is beyond the scope of this publication - again I am happy that this concern has been addressed.

Reviewer #2 (Remarks to the Author):

In their rebuttal the authors addressed all the questions raised during the review process. Unfortunately they have not been able to come up with new information that would have dispelled reviewer's concerns. Accordingly, some of the conclusions have had to be tuned down. Taking their word about the computational complexities involved I'd guess that is as much as it could be done.

About my specific comments I understand that it may be difficult to identify conservation of the interacting surfaces between RING1 proteins and the E2 ligase(s). Nevertheless, there are amino acids in the RING finger motif whose alteration render the (mammalian) monoubiquitylation module inert. In particular I53,D56 (human RING1B). These were identified, in fact, in structure models (PMC3160663, PMC4215650), despite the small size of the contact between E2 and RING1 proteins. I would have thought some speculation based on that. Indeed, I find these residues somehow conserved among the sequences in the added alignment, although I am sure the just by eye is not a rigorous enough argument. Likewise, amino acids critical for nucleosomal interaction K97R98 (in human RING1B) are highly conserved too. And yet, evidence for H2A monoubiquitylation is evolutionary restricted.

The addition of missing references and the widening of functional possibilities for PRC1 I just think them to be due for accuracy's sake.

Reviewer #3 (Remarks to the Author):

The Authors have addressed the reviewers comments and I think that this interesting work is ready for publication.

Reviewer #2 (Remarks to the Author):

In their rebuttal the authors addressed all the questions raised during the review process. Unfortunately they have not been able to come up with new information that would have dispelled reviewer's concerns. Accordingly, some of the conclusions have had to be tuned down. Taking their word about the computational complexities involved I'd guess that is as much as it could be done.

About my specific comments I understand that it may be difficult to identify conservation of the interacting surfaces between RING1 proteins and the E2 ligase(s). Nevertheless, there are amino acids in the RING finger motif whose alteration render the (mammalian) monoubiquitylation module inert. In particular I53,D56 (human RING1B). These were identified, in fact, in structure models (PMC3160663, PMC4215650), despite the small size of the contact between E2 and RING1 proteins. I would have thought some speculation based on that. Indeed, I find these residues somehow conserved among the sequences in the added alignment, although I am sure the just by eye is not a rigorous enough argument. Likewise, amino acids critical for nucleosomal interaction K97R98 (in human RING1B) are highly conserved too. And yet, evidence for H2A monoubiquitylation is evolutionary restricted.

Answer:

We thank the reviewer for highlighting the specific residues involved in the binding process, which greatly assisted us in reevaluating this aspect. To address this, we have introduced a new paragraph and an accompanying supplementary figure (SI Fig. S5) to provide further insights and considerations regarding the conservation and interaction of these critical amino acids.

Conservation of residues involved in H2AK199ub - but absence of H2AK199ub marks – suggests PRC2-independent functions of PRC1

Studies in human systems have revealed that H2AK119ub by PRC1 is particularly dependent on several key residues in the catalytic subunit RING1. Specifically, HsRING1B residues I53 and D56 are essential for stabilizing the interaction with the E2 ubiquitin ligase UbcH5c⁷⁷, and K97 and R98 mediate the interaction with nucleosomes⁷⁸. Both are essential for H2AK119ub in humans²². Of these key residues, I53, K97, and R98 are particularly well-conserved across RING1 orthologs in eukaryotes (**SI Fig.S5**), implying that the catalytic function of PRC1 could be conserved across species.

By contrast, D56 is only conserved in metazoan orthologs, suggesting that if PRC1 indeed has catalytic functions outside of metazoans, it is not dependent on D56 in taxa. Notably, RING1 orthologs in the plant *Arabidopsis thaliana* have a glycine at this position in the RING motif, and previous experimental works have shown that these orthologs do monoubiquitinate H2A in this species⁷⁹. Therefore, it appears that *Arabidopsis* RING1 is capable of ubiquitinating H2AK11 independent of the role D56 plays

in metazoans. More broadly, the equivalent hPTM of H2AK119ub has so far not been experimentally detected outside of holozoa and *Arabidopsis*⁶⁵. Thus, it remains unclear whether the conservation of I53, K97, and R98 across the breadth of eukaryotic diversity indeed correlates with widespread catalytic capacities of RING1 orthologs, or if those functions are truly limited to H2AK119ub in holozoa and *Arabidopsis*. This raises intriguing possibilities for the PRC1 complex to play PRC2-independent roles in gene regulation and chromatin modification in a wide range of organisms, beyond the traditionally associated functions in metazoans.